# TBX2 is a neuroblastoma core regulatory circuitry component enhancing MYCN/FOXM1 reactivation of DREAM targets

Bieke Decaesteker [1,2], Geertrui Denecker [1,2], Christophe Van Neste [1,2], Emmy M. Dolman[4], Wouter Van Loocke[1,2], Moritz Gartlgruber[5], Carolina Nunes [1,2], Fanny De Vloed [1,2], Pauline Depuydt [1,2], Karen Verboom [1,2], Dries Rombaut [1,2], Siebe Loontiens [1,2], Jolien De Wyn [1,2], Waleed M. Kholosy[4], Bianca Koopmans[4], Anke H.W. Essing [4], Carl Herrmann[6,7], Daniel Dreidax [5], Kaat Durinck[1,2], Dieter Deforce [2,8], Filip Van Nieuwerburgh [2,8], Anton Henssen [9,10,11], Rogier Versteeg [12], Valentina Boeva [3], Gudrun Schleiermacher [13], Johan van Nes [12], Pieter Mestdagh[1,2], Suzanne Vanhauwaert [1,2], Johannes H. Schulte[9], Frank Westermann[5], Jan J. Molenaar[4], Katleen De Preter[1,2] & Frank Speleman [1,2]

Chromosome 17q gains are almost invariably present in high-risk neuroblastoma cases. Here, we perform an integrative epigenomics search for dosage-sensitive transcription factors on 17q marked by H3K27ac defined super-enhancers and identify *TBX2* as top candidate gene. We show that *TBX2* is a constituent of the recently established core regulatory circuitry in neuroblastoma with features of a cell identity transcription factor, driving proliferation through activation of p21-DREAM repressed FOXM1 target genes. Combined *MYCN/TBX2* knockdown enforces cell growth arrest suggesting that *TBX2* enhances *MYCN* sustained activation of FOXM1 targets. Targeting transcriptional addiction by combined CDK7 and BET bromodomain inhibition shows synergistic effects on cell viability with strong repressive effects on CRC gene expression and p53 pathway response as well as several genes implicated in transcriptional regulation. In conclusion, we provide insight into the role of the *TBX2* CRC gene in transcriptional dependency of neuroblastoma cells warranting clinical trials using BET and CDK7 inhibitors.

[1] Center for Medical Genetics, Ghent University, Ghent 9000, Belgium. [2] Cancer Research Institute Ghent (CRIG), Ghent 9000, Belgium. [3] Institut Cochin, INSERM U1016, CNRS UMR 8104, Paris Descartes University UMR-S1016, Paris 75014, France. [4] Princess Máxima Center for Pediatric Oncology, Department of Translational Research, Utrecht 3584, The Netherlands. [5] Neuroblastoma Genomics, German Cancer Research Center (DKFZ), Heidelberg 69120, Germany. [6] Institute of Pharmacy and Molecular Biotechnology, and Bioquant Center, University of Heidelberg, Im Neuenheimer Feld 267, Heidelberg 69120, Germany. [7] Division of Theoretical Bioinformatics, German Cancer Research Center (DKFZ), Im Neuenheimer Feld 280, Heidelberg 69120, Germany. [8] Laboratory of Pharmaceutical Biotechnology, Ghent University, Ghent 9000, Belgium. [9] Department of Pediatric Oncology and Hematology, Charité-Universitätsmedizin Berlin, Berlin 13353, Germany. [10] German Cancer Consortium (DKTK), Heidelberg 69120, Germany. [11] Berlin Institute of Health, Berlin 13353, Germany. [12] Department of Oncogenomics, Academic Medical Center, Amsterdam 1105 AZ, Netherlands. [13] Institut Curie, PSL Research University, Equipe Labellisée Ligue contre le Cancer, Laboratory Recherche Translationnelle en Oncologie Pédiatrique (RTOP), Laboratoire "Gilles Thomas", Institut Curie, Department of Translational Research, Institut Curie, SIREDO: Care, Innovation and Research for Children, Adolescents and Young Adults with Cancer, Institut Curie, Paris 75248, France. These authors jointly supervised this work: Katleen De Preter, Frank Speleman. These authors equally contributed: Geertrui Denecker, Christophe Van Neste. Correspondence and requests for materials should be addressed to B.D. (email: bieke.decaesteker@ugent.be) or to F.S. (email: Franki.speleman@ugent.be)

Neuroblastoma (NB) is a cancer of the developing sympatho-adrenergic nervous system and is the most common malignancy diagnosed in children during their first years of life[1]. Sequencing revealed a relatively silent mutational landscape with only *ALK* activating mutations being identified in up to 10% of primary cases as well as de novo secondary or emerging subclonal ALK mutations in relapsed cases[2,3]. Further, in relapsed cases additional *RAS-MAPK* pathway driving mutations are enriched[4,5]. In contrast to mutations, DNA copy number alterations are remarkably recurrent in NB, including focal amplification of the *MYCN* oncogene in approximately half of the high-stage patients[6] and large 17q segmental gains occurring in the majority of both *MYCN* amplified and non-amplified high stage tumors[7–9]. The finding of recurrent gains of the syntenic human 17q region in MYCN driven NB mouse tumors further supports the putative functional importance of this genomic aberration in NB[10]. Investigating dosage-sensitive genes affected by recurrent copy number alterations can offer new insights into tumor biology as was illustrated in ependymoma where multiple dosage-affected genes, located within large chromosomal regions of recurrent gains and losses, were shown to act as oncogenes or tumor suppressors through installing a so-called cellular state driven through one or more altered cellular functions[11].

Given the recently proposed role of a core regulatory circuitry (CRC)[12] consisting of several super-enhancer (SE) marked[13] transcription factor constituents in NB[14–16], we decided to search for dosage-sensitive SE marked transcription factors encoding genes residing on chromosome 17q. The 'T-box 2 transcription factor' (*TBX2*), hitherto not reported to be implicated in NB, was prioritized as transcription factor with top-ranked SE score in NB cell lines and with expression levels highly correlated with survival outcome in NB tumors. *TBX2* is a member of the T-box family of transcription factors with an important role during embryogenesis and morphogenesis[17,18] and is overexpressed in several cancer entities including melanoma, breast, and pancreatic cancer[19–21]. The oncogenic effect of *TBX2* overexpression has been attributed to its role in proliferation as well as inducing epithelial-to-mesenchymal transition (EMT) and senescence bypass[22]. Based on integrated analysis of *TBX2* occupancy as determined by ChIP-sequencing and transcriptome analysis upon knockdown (KD), we propose *TBX2* as a novel bona fide constituent of the recently reported CRC in NB[14–16].

To investigate the role of *TBX2* in this CRC, functional analyses were performed showing the implication of TBX2 in cell cycle, proliferation, and downstream E2F-FOXM1 signaling. Finally, we demonstrate that combined pharmacological targeting of transcriptional addiction using a BET and CDK7 inhibitor, yields synergistic effects on *TBX2* downregulation leading to massive apoptosis.

## Results

### *TBX2* is a super-enhancer marked transcription factor on 17q.
CRCs consisting of SE marked master transcription factors were recently shown to be dysregulated in NB through MYCN-dependent transcriptional amplification[14,16] causing transcriptional addiction[23]. Given the highly recurrent chromosome 17q gain in high-risk human NBs and MYCN-driven mouse NBs, we hypothesized that one or more dosage-sensitive CRC transcription factors map to 17q thus rendering a selective advantage to tumors cells exhibiting 17q gain. To identify such transcription factors, we determined SE scores using the LILY algorithm[15] based on the intensity of H3K27ac marks in 26 NB cell lines with 17q gain, two non-malignant neural crest cell lines and the breast cancer cell line MCF-7 as non-embryonal control

(gene prioritization strategy is depicted in Fig. 1a, b and Supplementary Fig. 1a, b). We identified a total of 176 SE clusters on 17q of which six were present in at least 20 NB cell lines (Supplementary Fig. 1c). These six SE clusters are located in the vicinity of 86 candidate genes of interest, including 11 transcription factors[24], of which 5 are actively transcribed in NB cells, i.e. *TBX2, RARA, SP2, NFE2L1,* and *VEZF1*.

Next, we assessed the expression levels of these transcription factors in relation to patient survival in two independent NB tumor cohorts (GSE85047 $n = 276$, GSE62564 $n = 498$) and observed the strongest association with overall and progression-free survival for *TBX2*[25] (Fig. 1c and Supplementary Fig. 1d, Kaplan–Meier analysis). Moreover, TBX2 is marked by a SE in all investigated NB cell lines, but not in the human neural crest line (hNCC) and the MCF-7 breast cancer cell line (Fig. 1b and Supplementary Fig. 1b).

Of further interest, the highest expression levels for *TBX2* were observed in NB cell lines and primary tumors compared to other tumor entities, based on the online pan-cancer analysis in the CCLE database (cancer cell lines) and R2 platform (primary tumors and normal tissues) (Fig. 1d). *TBX2* expression levels were also high in normal embryonic tissues in keeping with the established role of *TBX2* in early development. Taken together, our data suggest a possible important role for *TBX2* as hitherto unrecognized transcriptional regulator in NB tumor development.

### 4C-seq defines *TBX2* promotor—super-enhancer interactions.
To provide further evidence for a functional role of the assigned SE for *TBX2* gene regulation, three different viewpoints residing in the SE region (20 kb and 260 kb up TSS) or the promoter site of *TBX2* (4.5 kb up TSS) were selected for 4C-sequencing in NB cell lines SK-N-AS and CLB-GA (Fig. 2a and Supplementary Fig. 2a). Reciprocal interactions were observed between the two viewpoints in the SE region and the promoter region, as well as interaction with a region more upstream of *TBX2* (400 kb up TSS). These results are in line with the proposed *TBX2* regulation according to the associated SE region as determined by H3K27ac mapping.

Of further interest, *TBX2* maps to the border of a topologically associated domain (TAD)[26] (Supplementary Fig. 2b) and has been associated with a bi-directionally transcribed topological anchor point (tap)RNA *TBX2-AS1*[27]. These positional conserved tapRNAs are located at chromatin loop anchor points and borders of TADs and show strong coordinated expression with their associated nearby protein-coding gene[28]. Indeed, we found a strong correlation between the expression levels of the tapRNA *TBX2-AS1* and *TBX2* in a large cohort of NB tumors ($n = 79$, Supplementary Fig. 2c). In addition to the potential regulatory connection of *TBX2-AS1* and *TBX2*, we also found strong correlation with expression levels of the *PPM1D* gene which maps within a 1.5 Mb distance from the *TBX2* locus (Supplementary Fig. 2d). In summary, the above findings support a physical interaction between the *TBX2* locus and its nearby SE and suggest that the proposed chromatin looping drives *TBX2* expression.

### *TBX2* is a copy-number affected dosage-sensitive gene.
Next, we investigated in more detail the genomic aberrations that account for the high *TBX2* expression in NB (Fig. 1d). We first analyzed the CCLE database and found that NB was the tumor entity exhibiting the most frequent gains for the *TBX2* locus, the highest expression and lowest methylation levels (Figs. 1d and 2b, c). Next, we assessed the effect of DNA copy number alterations on *TBX2* expression levels using an ANOVA analysis in the NRC tumor dataset ($n = 218$, GSE85047). Only in high-stage disease

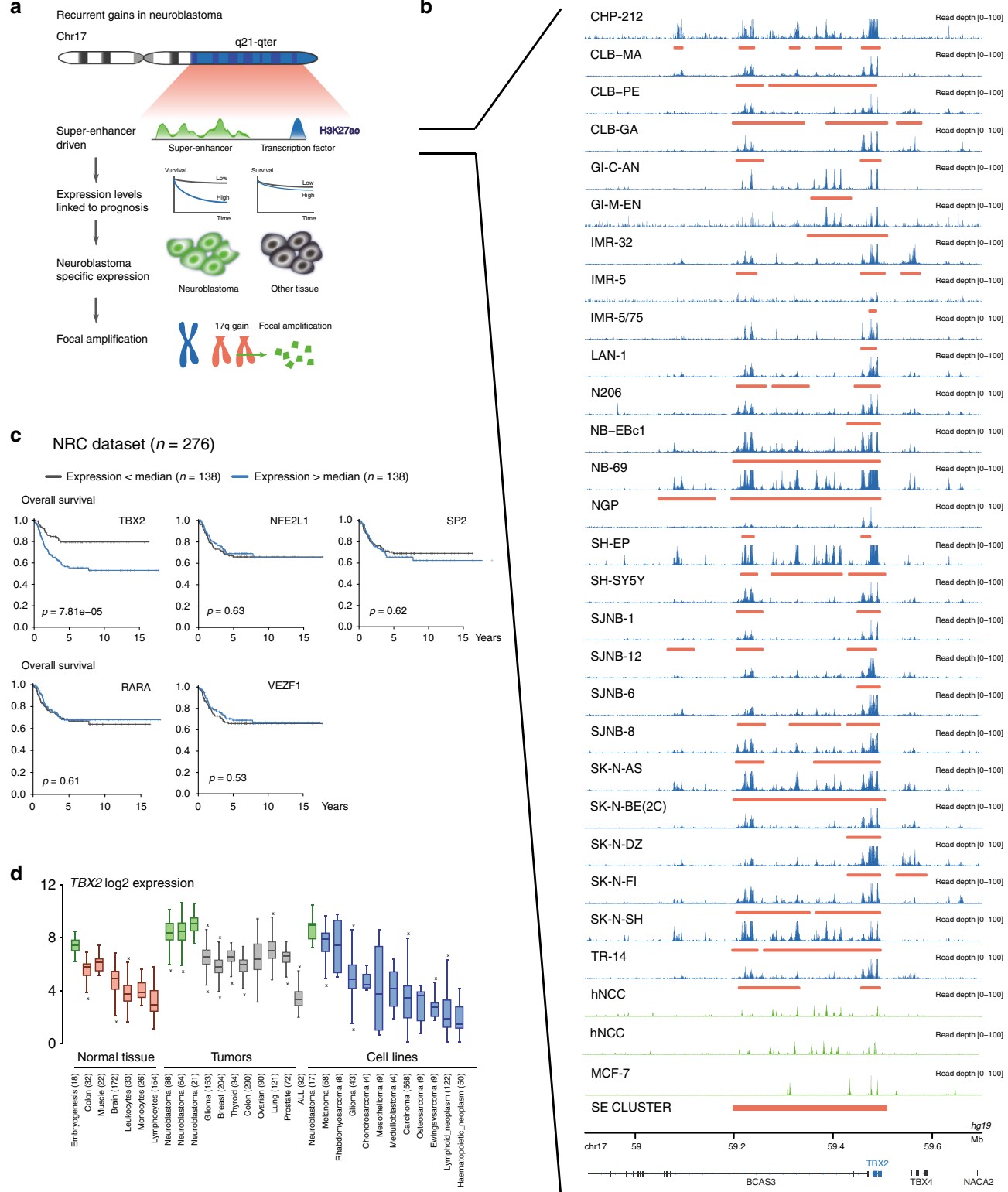

**Fig. 1** *TBX2* is a super-enhancer marked 17q transcription factor in NB. **a** Prioritization strategy to find SE-driven candidate oncogenes on chr17q. **b** H3K27ac activity in a region upstream of *TBX2* in 26 NB cell lines (blue), non-malignant neural crest cell lines (green) and the non-embryonal breast cancer cell line MCF-7 (green). The Lilly annotated SE regions are indicated in red. The cluster (out of 276 clusters on chr17q) containing the overlapping SE used in the prioritization process is annotated at the bottom. **c** Kaplan–Meier analysis (overall survival) of 276 neuroblastoma patients (NRC NB tumor cohort, GSE85047) with high or low expression (using median as cut-off) of the five prioritized candidate oncogenes (*TBX2, NFE2L1, SP2, RARA,* and *VEZF1*). **d** *TBX2* is highly expressed during embryogenesis as compared to normal adult tissue. *TBX2* is highly expressed in NB tumors and NB cell lines as compared to tumors or cell lines from other entities. Boxplots are drawn as a box, containing the 1st quartile up to the 3rd quartile of the data values. The median is represented as a line within the box. Whiskers represent the values of the outer two quartiles maximized at 1.5 times the size of the box. If one or more values outside of the whiskers are present, then this is indicated with a single mark 'x' next to the implicated whisker

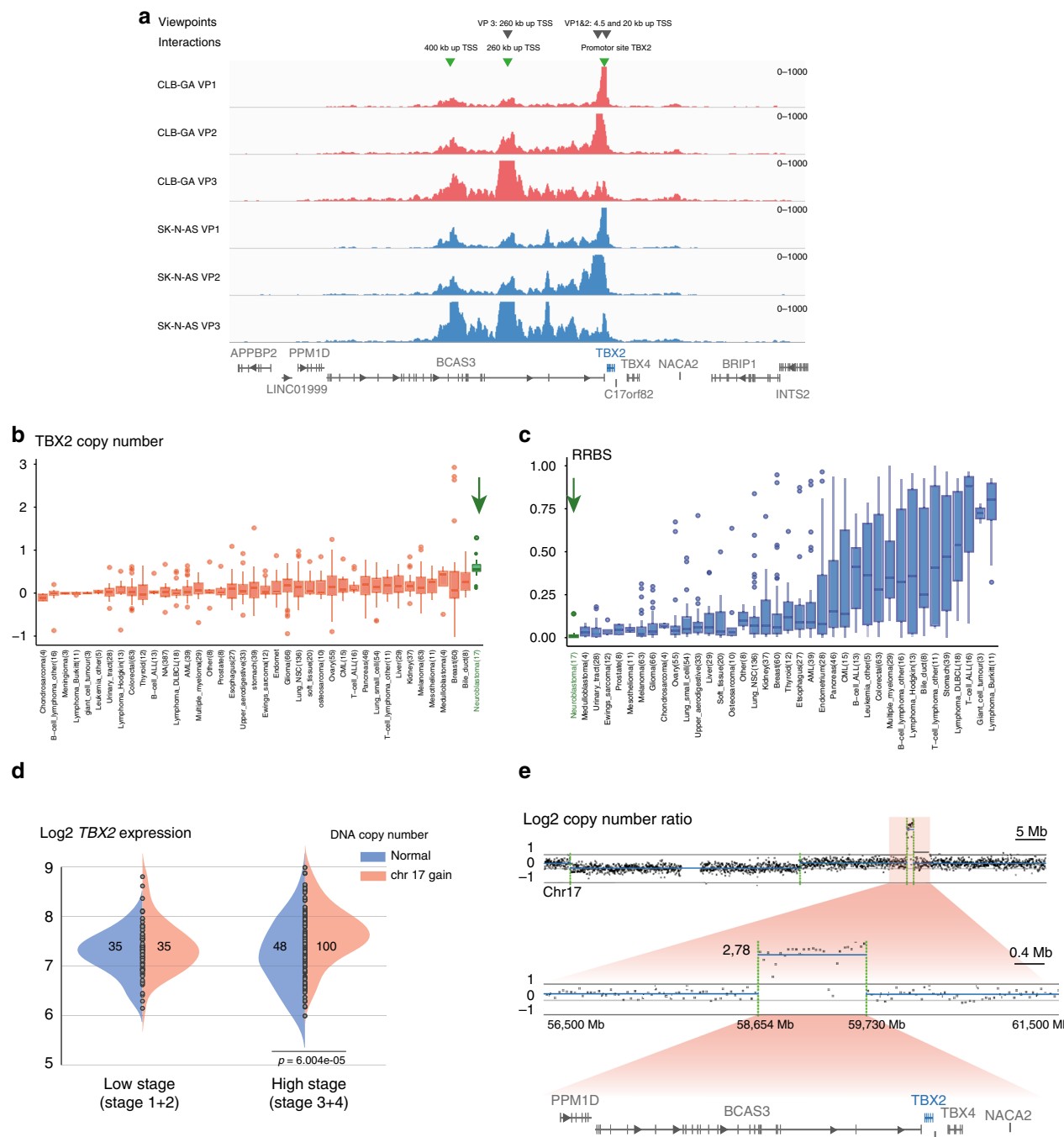

**Fig. 2** TBX2 is targeted by rare focal amplifications and marked by a SE. **a** 4C-seq analysis showing reciprocal interaction between the promotor site and SE upstream of *TBX2* in the NB CLB-GA and SK-N-AS cell lines using three different viewpoints. **b** Copy number ratio of the *TBX2* locus in a pan-cancer dataset (CCLE) with high copy number for *TBX2* in NB (green arrow) as compared to cell lines from other tumor entities. Boxplots are drawn as a box, containing the 1st quartile up to the 3rd quartile of the data values. The median is represented as a line within the box. Whiskers represent the values of the outer two quartiles maximized at 1.5 times the size of the box. If 1 or more values outside of the whiskers are present, then this is indicated with a single mark dot next to the implicated whisker. Number of samples for every entity is depicted below the boxplot. **c** Genome-wide methylation profile evaluated by reduced representation bisulfite sequencing (RRBS) of the *TBX2* locus in a pan-cancer dataset (CCLE). *TBX2* is covered by a low methylation profile in NB (green arrow) as compared to cell lines from other entities. Boxplots are drawn as a box, containing the 1st quartile up to the 3rd quartile of the data values. The median is represented as a line within the box. Whiskers represent the values of the outer two quartiles maximized at 1.5 times the size of the box. If one or more values outside of the whiskers are present, then this is indicated with a single mark dot next to the implicated whisker. Number of samples for every entity is depicted below the boxplot. **d** Significantly increased *TBX2* expression levels in cases with both high stage disease (stage 3 and 4) and increased *TBX2* copy number due to 17q gain (ANOVA, $n = 218$, $p = 6.004e{-}5$, chr17 gain log2 ratio > 0.3). **e** Log2 copy number ratio of a region on 17q with a focal amplification encompassing the protein-coding genes *PPM1D, BCAS3, TBX2, C17orf82, TBX4*, and *NACA2*, in a primary NB tumor case (58,654 Mb— 59,730 Mb—hg19—log2 ratio 2.78)

(stage 3 and 4), we observed significantly increased *TBX2* expression levels due to increased *TBX2* copy number ($p = 6.004e-5$, log2 ratio > 0.3) (Fig. 2d). We specifically looked for rare *TBX2* encompassing amplicons in a series of 556 high-risk NB cases[29] and detected a single *MYCN*-amplified case with an additional 1.076 Mb focal 17q23.2 amplification (Fig. 2e and Supplementary Fig. 2e) encompassing only six protein-coding genes including the transcription factors *TBX2* and *TBX4*. Of further note, a previously reported focal high level 1.8 Mb gain of a chromosome 17q23 segment in the NB cell line MP-N-TS also encompasses the *TBX2* locus[30,31]. Taken together, our data indicate that *TBX2* is a dosage-sensitive transcription factor affected by the common segmental 17q gains and rare amplification events in NB.

**TBX2 is a core regulatory circuitry constituent in NB**. To gain further insight into the *TBX2*-controlled regulatory network, we assessed TBX2 DNA occupancy by ChIP-sequencing and ATAC-sequencing in the NB cell line IMR-32. A total of 557 significant (adj.*P*.val < 0.05) TBX2 binding sites were identified and motif analysis confirmed enrichment for a TBX motif (AGGTGTGA, $p = 1e-41$), supporting the validity of our ChIP-seq data (Supplementary Data 1). In total, 81, 28, and 94% of TBX2 binding sites in IMR-32 respectively overlap H3K27ac, H3K4me3, and ATAC-sequencing peaks (Fisher test $p < 2.2e-16$, Fig. 3a, b), which confirms the binding of TBX2 to active promotor and enhancer regions. Moreover, respectively 41 and 30% of the TBX2 ChIP-seq peaks are found intergenic or are annotated to lncRNAs (Supplementary Fig. 3a, b), and 19% overlap with the SEs annotated in the cell line IMR-32 (Fisher test $p < 2.2e-16$, Supplementary Fig. 3c).

The recent reports on distinct CRCs in NB[14,15] prompted us to investigate the possible involvement of *TBX2*. In line with the recent finding of invasion of MYCN into non-canonical E-boxes at enhancers, motif analysis of TBX2-bound regions showed that the non-canonical MYC(N) E-box motif CANNTG was found to be highly enriched (Binomial test $p = 1e-63$) as well as motifs for GATA(1/2/3/4), PHOX2(A/B), HAND(1/2) and neuronal lineage-specific marker genes such as ASCL1, ISL1 and MEIS(1/2)[32,33] (Supplementary Data 1). We integrated the ChIP-seq tracks for TBX2 with those reported for GATA3[15,34], HAND2, PHOX2B[15], and MYCN (this study) in NB cell lines and observed overlap of TBX2 peak summits with binding sites of these CRC transcription factors (Fig. 3b), thus supporting the notion that TBX2 is indeed actively taking part in this CRC. Overlap of PHOX2B, HAND2, and GATA3 binding with the TBX2 peaks was predominantly observed in enhancer regions (Fig. 3b and Supplementary Fig. 3d). The integration of TBX2 into this CRC is further confirmed by the observation of auto-regulation by binding of the TBX2 transcription factor to its own SE constituent and binding of at least three CRC members including GATA3, HAND2 and PHOX2B within this SE constituent (Fig. 3c). In addition, TBX2 is binding the SE constituents of the other CRC members PHOX2B, GATA3, and HAND2, amongst others, as shown in Supplementary Fig. 3e. Finally, *TBX2* expression is positively correlated with *GATA3*, *HAND2*, and *PHOX2B* expression levels as well as with those of other potential CRC genes important in development in a NB tumor cohort ($n = 283$, Supplementary Fig. 3f). Taken together, our data suggest that *TBX2* is part of the recently described CRC together with *HAND2, GATA3*, and *PHOX2B*.

**TBX2 controls E2F-FOXM1 driven cell cycle and proliferation**. To unravel the role of *TBX2* within the CRC in NB cells, we performed *TBX2* KD with two shRNAs and a non-targeting

control and subsequent gene expression profiling in the NB cell line IMR-5/75 (Supplementary Fig. 4a). A total of 1055 and 1326 genes were differentially down and upregulated, respectively (adj. *p*.val < 0.05, Supplementary Data 2), including the upregulated gene *CDKN1A*, which is a known target gene repressed by TBX2[35]. Gene set enrichment analysis (GSEA) on the down-regulated genes upon *TBX2* KD in IMR-5/75 cells revealed enrichment (FDR < 0.01) for the hallmark and gene ontology gene sets involved in cell cycle including G2/M checkpoint, E2F, MYC(N) targets, mitosis, and DNA replication (Fig. 4a, Supplementary Data 3) and enrichment was shown for TP53 pathway among the upregulated genes. Using iRegulon, designed to detect transcription factors, targets and motifs/tracks from a set of genes[24], (http://iregulon.aertslab.org/), we identified motif enrichment (FDR < 0.01) for FOXM1, E2F, and E2F binding partners TFDP1/TFDP3 in nearly half of all downregulated genes (Fig. 4b, Supplementary Data 4). Interestingly, an E2F motif was also enriched in the TBX2 binding sites in IMR-32 cells (Supplementary Data 1). The role of E2F and the MuvB core component FOXM1 was further supported by significant enrichment for published gene sets containing FOXM1, DREAM, E2F, and CCND1/CDK4 (the latter is known to phosphorylate FOXM1) activity/target genes[36] (Fig. 4c). In contrast, almost half of upregulated genes upon *TBX2* KD showed enrichment for a motif with high similarity to the MuvB core component MYBL2, REST (transcriptional repressor implicated in neuronal differentiation[37]) and EP300 (histone acetyltransferase) ChIP binding sites (Fig. 4b, Supplementary Data 4). In addition to FOXM1 target genes, *FOXM1* itself is also downregulated upon *TBX2* KD in the IMR-5/75 and CLB-GA cell lines (Fig. 4d).

Using a similar approach as for *MYCN* amplified IMR-5/75 cells, we also investigated the *TBX2* transcriptional regulated network in the non-MYCN amplified CLB-GA NB cells and observed similar downstream targets and enriched genes sets as those observed in IMR-5/75 upon *TBX2* KD (Supplementary Fig. 4a–d, Supplementary Data 3 and 5), with the *E2F/FOXM1* axis being most prominent.

To further validate these results, we performed correlation analysis in publicly available transcriptome data of 283 primary NB tumors (NRC, GSE85047). Supporting the *TBX2* KD data described above, GSEA showed enrichment for cell cycle, DNA repair and DNA replication as well as chromatin architecture among the genes positively correlated with *TBX2* expression levels in the NB tumor data set (FDR < 0.01, $R > 4$, Fig. 4e). In addition, we found a significant positive correlation for expression levels of *TBX2* versus *FOXM1*, *E2F* core members, and other DREAM complex members (Supplementary Fig. 4e). Furthermore, the *TBX2* KD signature score (generated in both CLB-GA and IMR-5/75 cells) was negatively correlated with the expression of these DREAM complex members (Supplementary Fig. 4e, f).

As these data collectively suggest a role for TBX2 in control of a FOXM1/E2F-driven gene regulatory network driving proliferation, we explored the phenotypical effects of *TBX2* KD in NB cells following prolonged lentiviral short hairpin RNA mediated KD of *TBX2* in IMR-32, CLB-GA and SK-N-AS NB cell lines with high TBX2 expression. KD of *TBX2* with four different hairpins for 5 days (Supplementary Fig. 4a) leads to a decrease in colony formation capacity (Fig. 5a and Supplementary Fig. 5a) and proliferation as measured with time-lapse microscopy (every 2–3 h) (Fig. 5b). Furthermore, we also show a significant G1 growth arrest in the CLB-GA cell line upon *TBX2* KD (Fig. 5c) in keeping with the above-reported effect of *TBX2* KD on cell cycle genes. Importantly, no effect on proliferation or colony formation capacity was detected in the *TBX2* non-expressing SH-EP cell line, indicating limited off-target effects with these four hairpins (Supplementary Fig. 5b). In summary, we have shown that TBX2

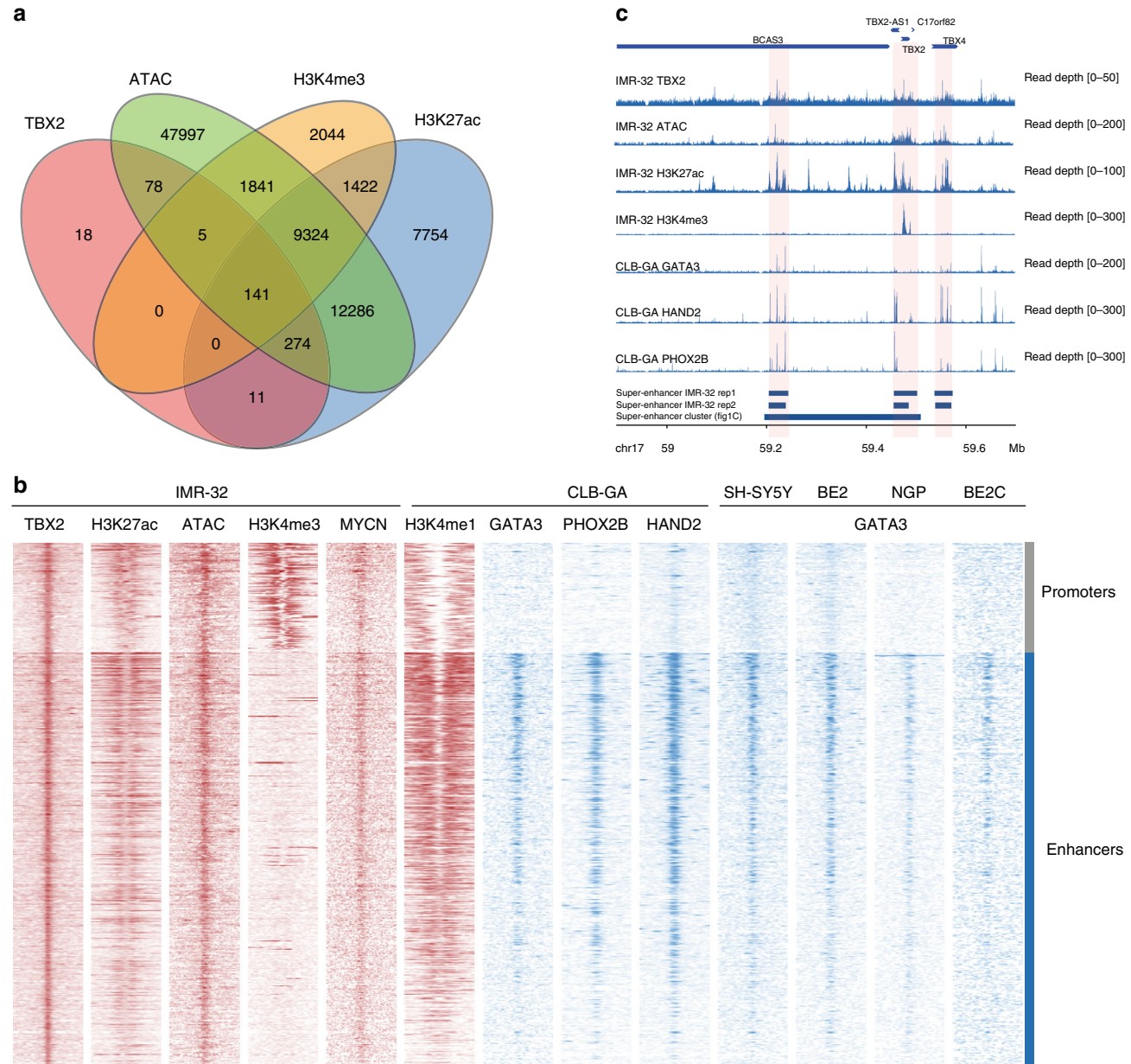

**Fig. 3** *TBX2* is part of a core-regulatory circuitry. **a** Significant (Fisher test *p*-value < 2.2e−16) overlap (min. overlap = 20 bp) of the TBX2 (qval < 0.05), H3K27ac (qval < 0.05), H3K4me3 (qval < 0.05) ChIP-seq and ATAC (qval < 10ˆ−6) peaks in the IMR-32 cell line. **b** Heatmap profiles −4 kb and +4 kb around the summit of TBX2 peaks. The heatmaps represent the ChIP-seq overlap of TBX2, H3K27ac, ATAC, H3K4me3, and MYCN peaks in the IMR-32 cell line, the histone mark H3K4me1 and the CRC genes *GATA3*, *PHOX2B*, and *HAND2* in the CLB-GA cell line, and *GATA3* in the SH-SY5Y, BE2, NGP, and BE2C cell lines, grouped for promoters or enhancers (homer annotation), and ranked according to the sums of the ChIP-seq peak scores across all ChIP-seq peaks in the heatmap. **c** *TBX2* region (59.1 Mb–59.65 Mb) with the TBX2, ATAC, H3K27ac, and H3K4me3 ChIP profiles in IMR-32 and GATA3, HAND2 and PHOX2B ChIP profiles in CLB-GA. At the bottom, the Lilly called SEs are annotated for the two replicates in IMR-32 as well as the cluster of SEs used for the prioritization (Fig. 1c)

is implicated in cell cycle and proliferation through regulating an E2F-FOXM1 driven cellular state.

**Combined TBX2-MYCN signaling targets the FOXM1/E2F network.** While CRC transcription factors bind their own enhancers as well as those of the other CRC partners, the individual contribution of each factor to the malignant phenotype of the cancer cells is largely unexplored. A recent report suggested that TWIST and HAND2 would cooperate with MYCN through binding of enhancers of a set of developmental genes[16]. In view of

these findings, we also decided to further study the TBX2–MYCN interrelationship. We first obtained indirect evidence for a functional relationship between MYCN and TBX2 from several datasets: (1) significant enrichment for publicly available MYC (N) target/signature gene sets in the *MYCN* amplified IMR-5/75 and *MYCN* non-amplified cell line CLB-GA upon *TBX2* KD (Figs. 4a, 6a and Supplementary Fig. 4d); (2) significant correlation of *TBX2* and *MYCN* mRNA and protein expression in NB tumors (Supplementary Fig. 3f) and NB cell lines (Supplementary Fig. 6a), respectively; (3) correlation between MYCN activity and TBX2 shRNA signature scores (established in both IMR5/75 and

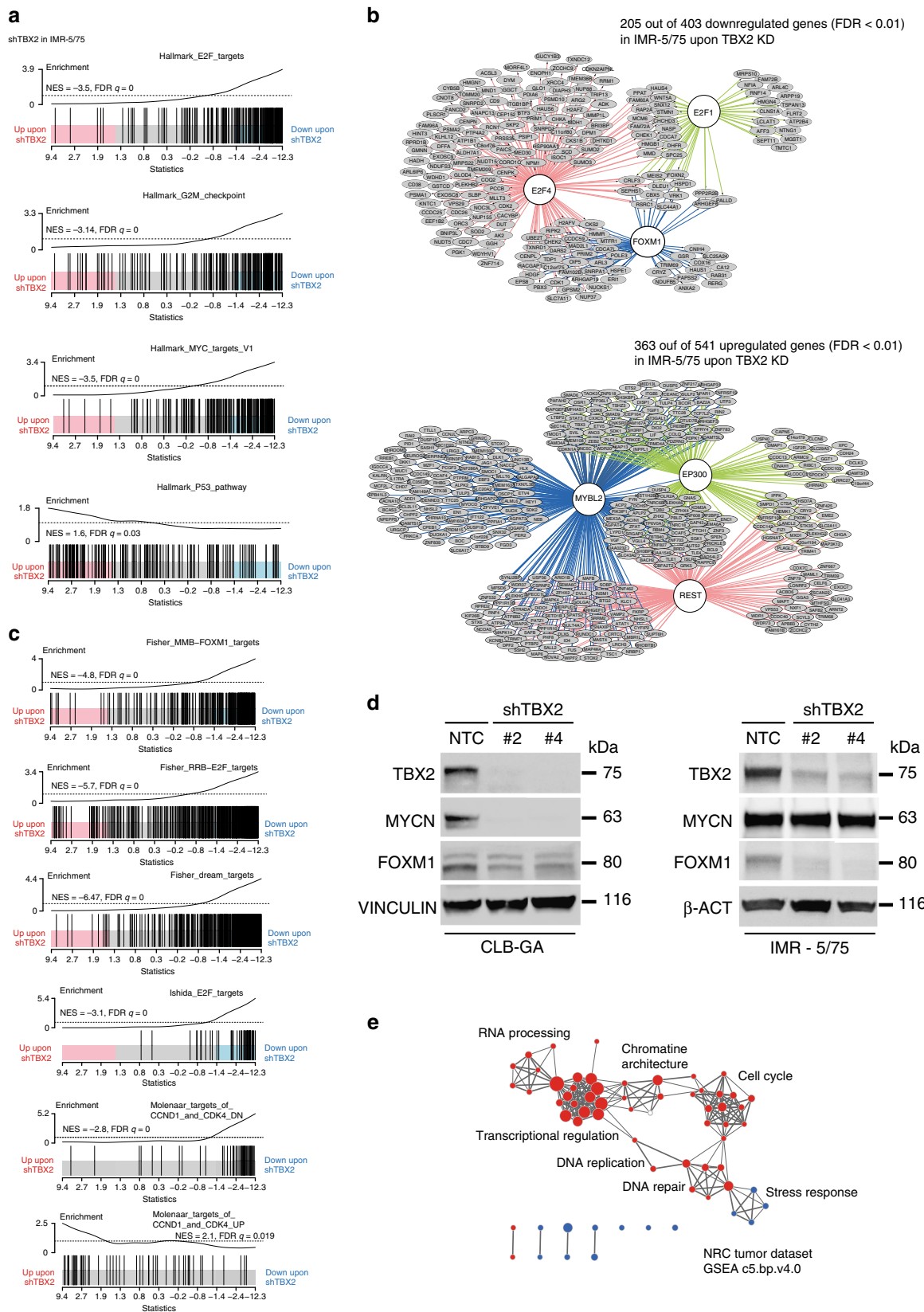

CLBGA) in NB tumors (CLB-GA: Spearman Correlation $p = 1.28e{-}83$, $R = -0.684$, IMR-5/75: Spearman Correlation $p = 7.91e{-}26$, $R = -0.576$) (Supplementary Fig. 6b) and NB cell lines (CLB-GA: Spearman Correlation $p = 0.00179$, $R = -0.564$, Spearman Correlation IMR-5/75: $p = 6.67e{-}08$, $R = -0.825$); (4)

62% of differentially expressed genes upon *TBX2* KD (FDR < 0.05, $n = 2381$) in IMR5/75 overlapped (Fisher test, $p = 0.006$) with the differentially expressed genes upon (dox inducible) *MYCN* KD in the same cell line (FDR < 0.05, $n = 4409$), indicating that these transcription factors co-regulate the same gene

**Fig. 4** TBX2 controls a FOXM1/E2F gene regulatory network. **a** Top enriched MsigDB hallmark genesets among the downregulated genes upon sh*TBX2* in IMR-5/75 (FD < 0.01) and the upregulated genes (FDR < 0.01) upon sh*TBX2* in IMR-5/75. Normalized enrichment score (NES) and false discovery rate (FDR) is depicted on the barcode plot. **b** iRegulon motif search for the downregulated and upregulated genes (FDR < 0.01) upon *TBX2* knockdown in IMR-5/75. Genes connected to a gene in a white circle do have a motif enrichment or ChIP-seq binding for the respective gene in the circle. **c** GSEA results for genesets from literature (FOXM1 and E2F targets, and the downregulated genes upon *CCND1* and *CDK4* knockdown) which are enriched in the *TBX2* downregulated genes in IMR-5/75. Normalized enrichment score (NES) and false discovery rate (FDR) is depicted on the barcode plot. **d** Western blot of TBX2, MYCN, and FOXM1 levels upon *TBX2* knockdown, in the CLBGA *MYCN* single copy and IMR-5/75 *MYCN* amplified cell lines. **e** Clustering of genesets (MsigDB c5.bp.v4.0) correlated with *TBX2* expression levels in the NRC tumor cohort (*n* = 283, GSE85047). Red nodes represent the gene sets positively correlated with *TBX2* expression (FDR < 0.01, *R* > 4), blue nodes represent gene sets negatively correlated with *TBX2* expression (FDR < 0.01, *R* < −0.3). Size of nodes depicts the size of the gene sets. Nodes that are clustered represent gene sets with the same or similar functional indication

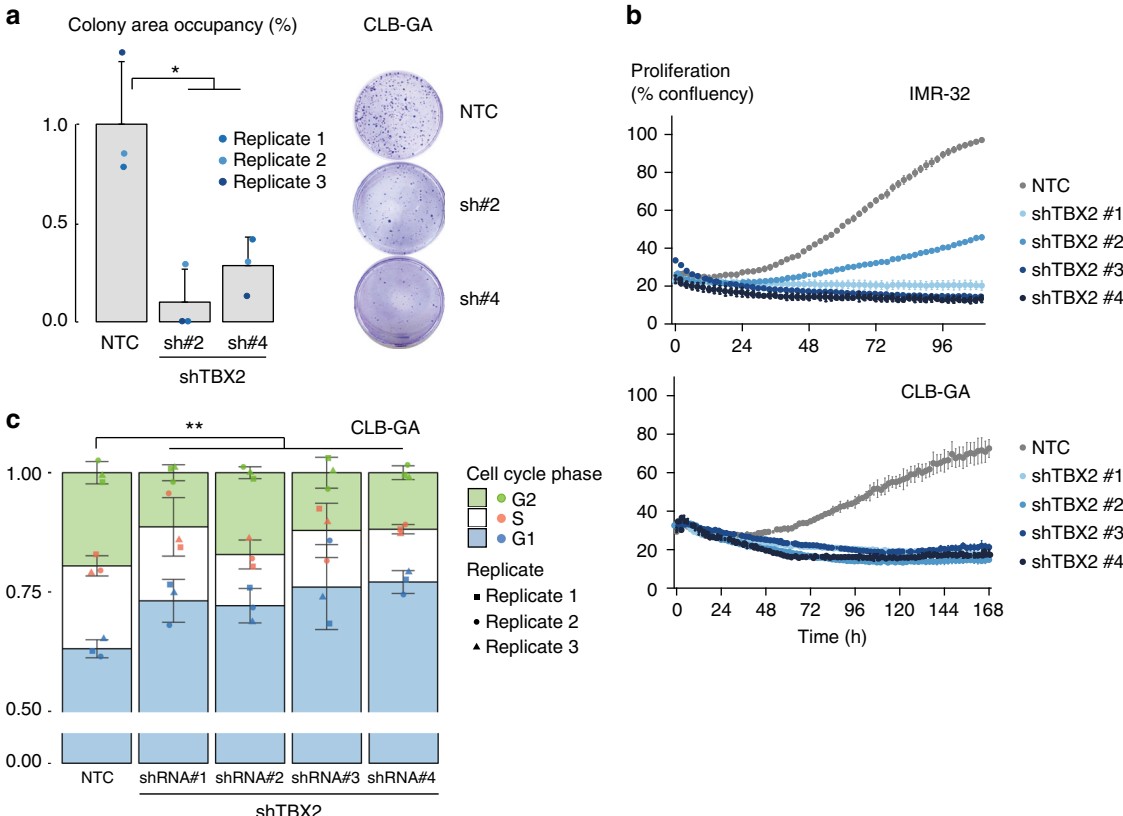

**Fig. 5** *TBX2* is a cell dependency gene in neuroblastoma. **a** Significant reduction in colony area occupancy (%) upon *TBX2* knockdown as compared to the non-targeting control (NTC) in CLB-GA cells. **b** Reduction of proliferation in IMR-32 and CLB-GA cells upon *TBX2* knockdown measured with time-lapse microscopy (every 2–3 h) using the IncuCyte device (one out of three biological replicates is shown). **c** Cell cycle analysis upon *TBX2* knockdown in the CLB-GA cell line. Significant induction of G1 cell cycle phase arrest upon *TBX2* knockdown (statistical test is based on G1 phase percentage). Data-points were mean-centered (a,c) and error bars represent the s.d. of three biological (a,c) or five technical (b) replicates for every cell line. Mann–Whitney test *p < 0.05, **p < 0.01

sets and finally (5) significant dynamical downregulation of the sh*TBX2* signature was noted during Tg(TH-MYCN) driven NB formation in transgenic mice, at one, two and six weeks after birth[38] (Supplementary Fig. 6c).

To further experimentally explore this presumed cooperation between *TBX2* and *MYCN*, we assessed the effects of *TBX2* KD in the presence of high versus low *MYCN* levels. Using this approach, we observed a stronger decrease in cell proliferation and increased G1-phase arrest when combining *TBX2* and *MYCN* KD together (Fig. 6b, c). Next, we performed RNA-sequencing to further explore the transcriptional effects of individual versus combined *MYCN* and *TBX2* KD and confirmed synergistic effects on expression levels of gene sets implicated in cell cycle and the DREAM-E2F-FOXM1 complex, as well as publicly

available MYCN signatures (Fig. 6d). Using iRegulon analysis on the enforced affected genes, motif enrichment of DREAM complex core members, such as FOXM1, E2F4, and MYBL2 was observed in the additively downregulated genes, while ChIP-seq targets for EP300 and NANOG (public datasets) were enriched in the upregulated genes (Supplementary Fig. 6d, Supplementary Data 4).

MYCN protein levels were decreased upon *TBX2* KD in the MYCN non-amplified CLB-GA cell line, while in the *MYCN* amplified IMR-5/75 cell line this was not the case (Fig. 4d). While we show that both transcription factors have an effect on each other's transcriptional activity, the reciprocal effect on expression seems to be complex and might be regulated via feedback loops. The study of these interactions is beyond the scope of this study.

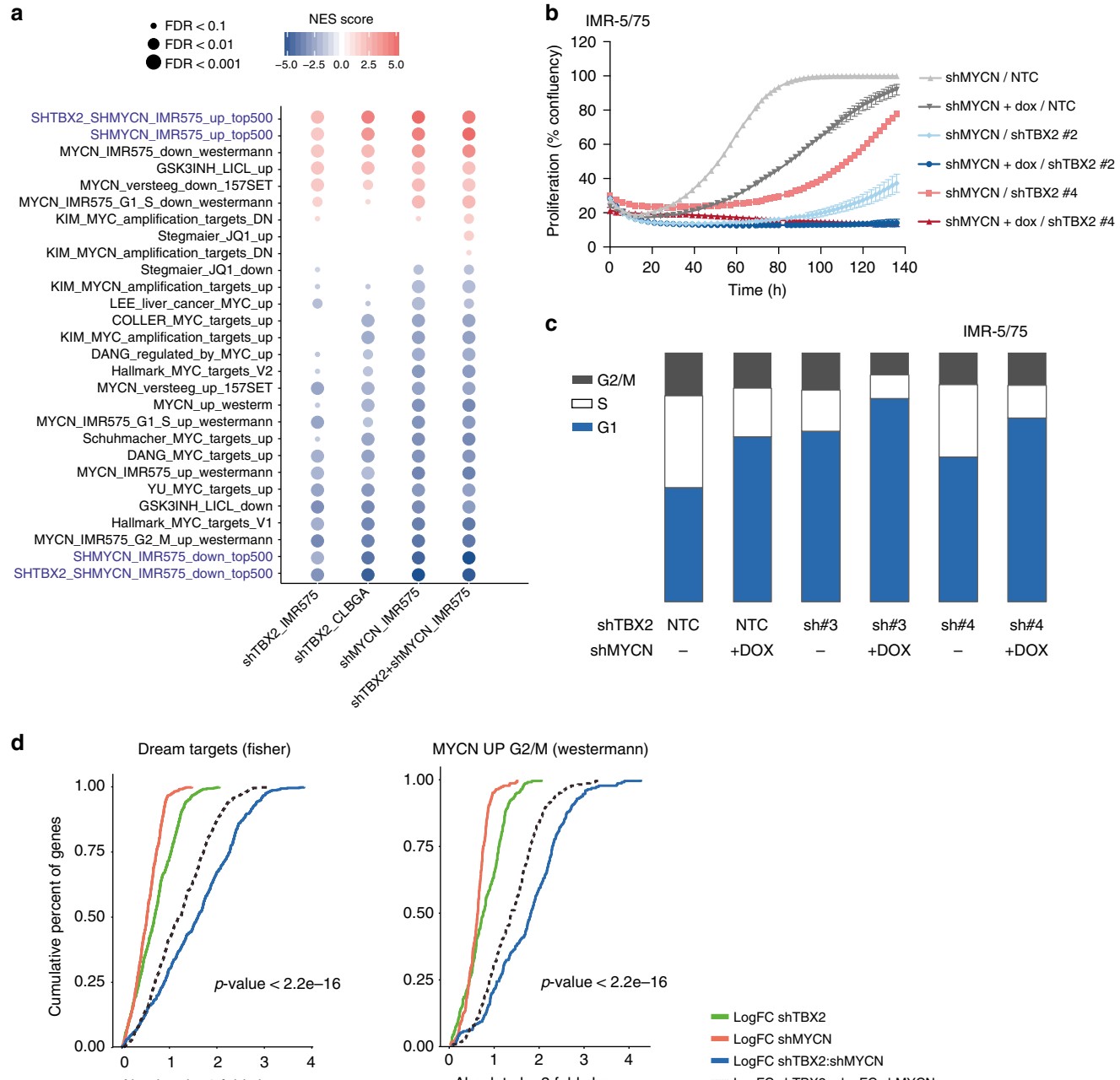

**Fig. 6** Combined TBX2-MYCN signaling targets the FOXM1/E2F network. **a** Heatmap showing GSEA enrichment scores (with FDR < 0.1) for an in house compiled gene set collection containing all MYC target genesets from the "Hallmark V5.1" catalog (MsigDB) as well as publically available MYC(N) activity or target signatures upon knockdown of *TBX2* in IMR-5/75, CLB-GA, knockdown of *MYCN* in IMR-5/75 or the combination of *TBX2* and *MYCN* knockdown in IMR-5/75. Size of the circles indicate FDR value and the color indicates the NES score. Positive values (red) point at enrichment among the upregulated genes while negative values (blue) indicate enrichment among the downregulated genes. **b** Synergistic reduction of cell proliferation upon combined knockdown of *MYCN* and *TBX2* in the IMR-5/75 cell line measured with time-lapse microscopy (every 2–3 h) using the IncuCyte device (one out of three biological replicates is shown). Error bars represent the s.d. of five technical replicates. Bliss independence score was used to calculate potential synergism at 120 h for both the sh*TBX2* #2 and #4 compared with the double *MYCN* and *TBX2* knockdown. Excess over bliss score for sh*TBX2* #2 is 0.092, excess over bliss score for sh*TBX2* #4 is 0.342. Excess over Bliss scores > 0 indicates synergy, < 0 indicates antagonism. **c** Combined *TBX2* and *MYCN* knockdown in the IMR-5/75 cell line results in an enforced G1-cell phase arrest with almost no cells in S-phase or G2/M phase left (one replicate out of 3 shown). **d** Cumulative distribution plot of the absolute log fold changes of sh*TBX2*, sh*MYCN* or sh*TBX2*:sh*MYCN* up- and downregulated genes. Dashed line represents the sum of the log fold changes of the sh*TBX2* and sh*MYCN* condition for the respective genes in the geneset. The difference between the blue line and the dashed line indicates synergism effect on logFC for the genes in the genesets implicated in cell cycle and *MYCN* regulation (two-sided paired *t*-test: *p* < 2.2e−16)

Although *FOXM1* is not regulated directly by TBX2, MYCN did show binding to the *FOXM1* promoter in several NB cell lines (data not shown).

In summary, our data support the cooperation of *TBX2* and *MYCN* regulating the NB driven proliferative cellular state mediated by *FOXM1*.

**Drugging transcriptional addiction to TBX2 and CRC genes**. To test the possibility to drug the transcriptional addiction of NB cells to the highly expressed *TBX2* gene and CRC genes, we combined the CDK7 inhibitor THZ1, which was previously shown to affect transcription of lineage-dependency genes in NB[39], with the bromo-domain inhibitor JQ1, which causes transcriptional repression of SE associated oncogenes[40]. We observed a significant synergistic effect on cell viability upon evaluation of a concentration range of JQ1 (5.1 nM–33.3 μM) and THZ1 (0.051 nM–0.333 μM) in 5 NB cell lines (i.e. 2 *MYCN* amplified cell lines Kelly and SK-N-BE(2c) and 3 *MYCN* non-amplified cell lines CLB-GA, SH-SY5Y, and SK-N-AS) (Fig. 7a), after 72 h treatment. In addition, we expanded this analysis for two primary patient-derived tumor cell lines (one *MYCN* amplified and one *MYCN* non-amplified organoid grown in stem cell medium) for which we tested viability for drug combinations after 5 days of treatment and observed even stronger synergism (Fig. 7b). Based on these findings, we selected concentrations of 35 nM for THZ1 and 1 μM for JQ1 to evaluate treatment response over time in eight cell lines (cell lines above and IMR-32 and IMR-5/75), and observed an increasing synergistic effect on cell proliferation and apoptosis in all tested NB cell lines, while MCF-7 remained unaffected (Fig. 7c, Supplementary Fig. 7a, b). Given that TBX2 was previously implicated in HDAC1 controlled repression of *CDKN1A* expression and cell cycle arrest in different cancer types[20] and the strong observed effects of HDAC inhibitors in combination with other anti-cancer drugs[41], we also decided to combine HDAC1 inhibitor Panobinostat together with the CDK7 inhibitor THZ1 (Supplementary Fig. 7c). We observed a significantly synergistic effect over time on cell proliferation and apoptosis albeit only in four out of eight NB cell lines (Supplementary Fig. 7d, e).

Based on these data, we identify combined BET and CDK7 inhibition as a potent synergistic drug combination to target NB cells. To gain deeper insight into the molecular basis of the observed JQ1/THZ1 drug synergism, we performed gene expression profiling after a 10 h treatment with 35 nM THZ1 and 1 μM JQ1 in cell line IMR-5/75. Treatment with the single and combined compounds resulted in a more than two-fold reduction in steady-state mRNA levels with 4.3, 3.1 and 7.9% respectively as compared to the control (Supplementary Fig. 8a). First, we evaluated synergistic effect of combined JQ1 and THZ1 treatment on transcriptional dependency of the NB cells for *TBX2* and other CRC genes and confirmed strong reduction in expression levels (Supplementary Fig. 8b). More specifically, all predefined CRC genes in NB cell lines with high *TBX2* expression[16] or in the (nor)adrenergic module[14,15] were significantly downregulated upon combination treatment (Fig. 8a, Supplementary Fig. 8c). Downregulation of *TBX2* expression levels was confirmed by qPCR analysis after single compound treatment while combination of THZ1 and JQ1 treatment yielded further downregulation in the high *TBX2* expressing IMR-32, IMR5-75 and Kelly cells (Fig. 8b). Furthermore, *PHOX2B*, *FOXM1*, and the *LIN28B* (implicated in a *MYCN-LIN28B* regulatory axis)[42] mRNA and protein expression levels were also strongly affected (Fig. 8b, c, Supplementary Fig. 8d).

In addition to the CRC genes, we also observed dramatic altered gene expression patterns for *FOXM1/E2F*/DREAM complex core genes upon combination treatment. While these effects are robust, the direct of regulation does not allow an unequivocal interpretation of the effects of drug synergism. For example, *MYBL2* is strongly induced and acts, together with FOXM1 as an activator of genes driving cell cycle progression. One possible explanation is the existence of a feedback loop as reported[43] (Fig. 8d, Supplementary Fig. 8e). In addition to the observed effects on CRC genes and *FOXM1/E2F*/DREAM complex core genes, the drug synergism also effects TP53 pathway response (Fig. 8e), in keeping which the finding that TP53 activation may sensitize transcriptionally addicted cancer cells to THZ1 inhibition[44].

Further scrutinizing of the top differentially and synergistically downregulated genes (Supplementary Data 6) contains several important regulators of transcription including several chromatin remodelers such as *BPTF*, *IWS1*, and *INO80* which can be assumed to be implicated in the drug synergism. In the top upregulated genes upon treatment with JQ1 ($p = 0.05$), THZ1 ($p = 0.05$) and the combination ($p = 0.01$), we noticed *HEXIM1*, a presumed tumor suppressor which forms an inhibitory complex with P-TEFb and implicated in cell cycle progression and TP53 response[45], the two major phenotypic effects observed upon *TBX2* KD (Supplementary Fig 8f)).

Taken together, we propose that the *MYCN-TBX2* CRC represents an important novel therapeutic vulnerability for high-risk NB, warranting future clinical trials to assess available BET and CDK7 inhibitors.

**Discussion**
The high-risk NB genome is dominated by DNA copy number alterations, the most prominent being *MYCN* amplification occurring in roughly half of these cases. We and others have previously shown that chromosome 17q gain is the most frequent alteration in both *MYCN* amplified and non-amplified high-risk NB[7–9]. In addition, the syntenic region in *MYCN*-driven mouse NB also undergoes copy number increase[10]. Recently, SE marked master transcription factors were identified in NB that co-occupy most enhancers and form an auto-regulatory loop what has been called a CRC[14–16].

We hypothesized that expression levels and activity of one or more CRC constituents could be affected through chromosome 17q copy number gains. To test this hypothesis, we ranked transcription factors on chromosome 17q based on H3K27ac mark and patient survival, and identified *TBX2* as top candidate. *TBX2*, which has thus far not been studied in NB, is most highly expressed in this tumor entity as compared to other tumor entities and also strongly upregulated in mouse neural crest-derived MYCN overexpressing NB[46]. In addition to the common *MYCN* amplification, rare amplicons have been shown to impact on activity of important NB genes as has been illustrated for *ALK* and *LIN28B*[47,48]. We also reported recently that such rare amplicons have strong negative impact on survival in keeping with an important role of target genes in these rare amplicons in the general NB tumor biology and clinical behavior[49]. In the light of these observations, the finding of *TBX2* amplification in two NB cases further supports the biological relevance of this gene.

Analysis of the present *TBX2* and *MYCN* occupancy data and available data for CRC constituents such as *GATA3*, *PHOX2B*, and *HAND2* provided compelling evidence that *TBX2* also functions as a CRC gene in NB. Dysregulation of these CRCs has been proposed to result from oncogenic master transcription factors, altered transcription of one or more CRC downstream signaling genes or a more global downstream perturbation due to invasion of a transcriptional amplifier[50]. While our current understanding of the specific mode of action and contribution of

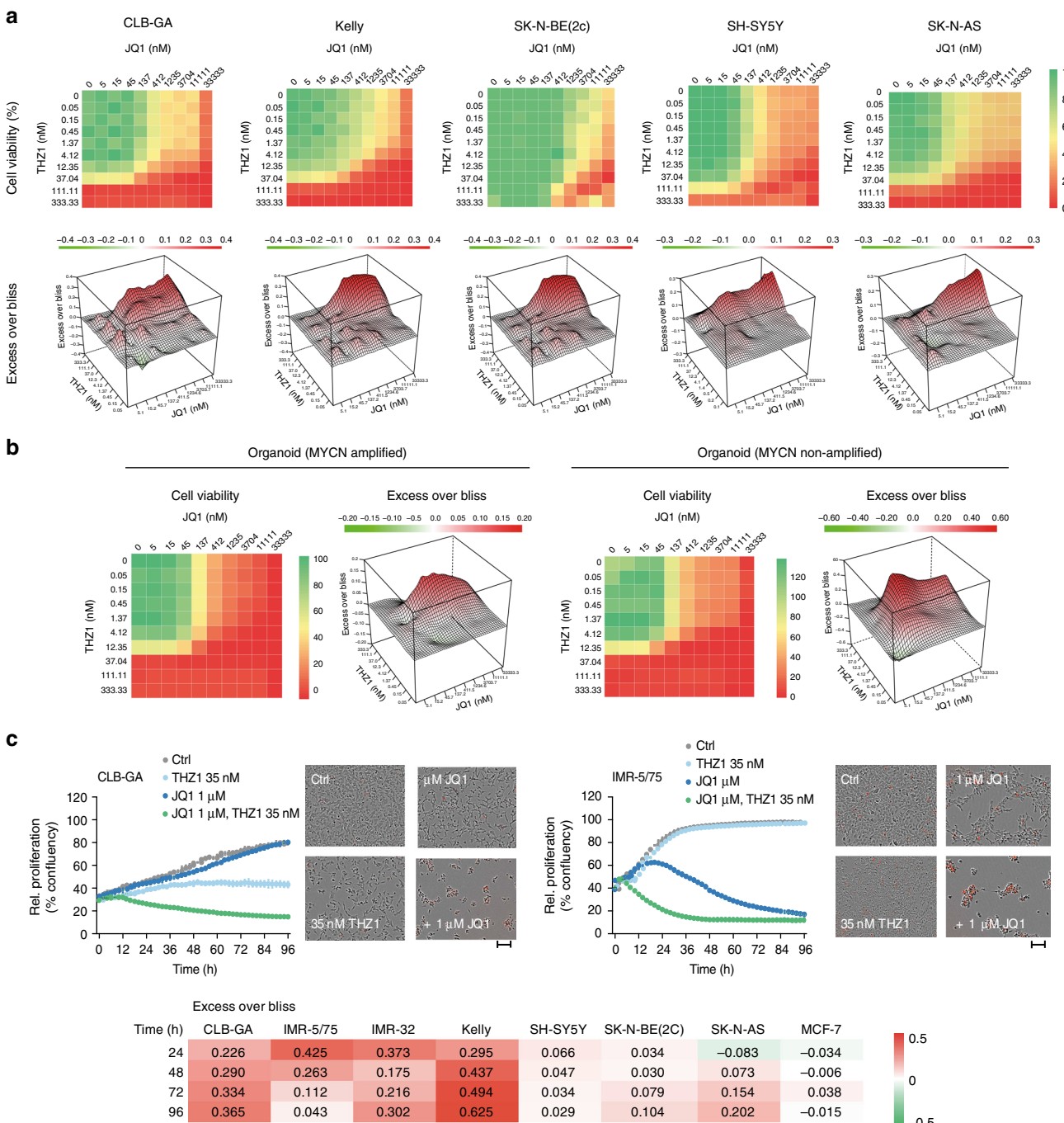

**Fig. 7** Combined CDK7-BET inhibition as a novel therapeutic approach. **a** Heatmap with the percentage of cell viability for serial dilutions of JQ1 (5.1 nM–33.3 μM) in combination with THZ1 (0.051 nM–0.333 μM) in Kelly, SK-N-BE(2c), SK-N-AS, IMR-32, CLB-GA, and SHS-Y5Y), 72 h upon treatment; and respective 3D representation of Excess over Bliss scores. Excess over Bliss scores >0 indicates drug synergy, <0 indicates drug antagonism. Data points in the screen represent mean of two technical replicates. **b** Heatmap with the percentage of cell viability for serial dilutions of JQ1 (5.1 nM–33.3 μM) in combination with THZ1 (0.051 nM–0.333 μM) in two organoids (one *MYCN* amplified and one *MYCN* non-amplified, 5 days upon treatment); and respective 3D representation of Excess over Bliss scores. Excess over Bliss scores >0 indicates drug synergy, <0 indicates drug antagonism. Data points in the screen represent mean of two technical replicates. **c** Synergistic effect on proliferation (% confluency) over time for the cell lines CLB-GA and IMR-5/75 upon treatment with 35 nM THZ1 and 1 μM JQ1. One biological replicate out of three is shown, error bars represent the s.d. of the three technical replicates. Microscopic pictures of the cells 60 h upon treatment are depicted at the right (scale bar is 100 μm). Red staining of the cells indicates apoptosis (AnnexinV positivity). Bliss score was used to calculate potential synergism for 7 NB cell lines and the negative control MCF-7 breast cancer cell line. Excess over Bliss scores >0 indicates drug synergy, <0 indicates drug antagonism

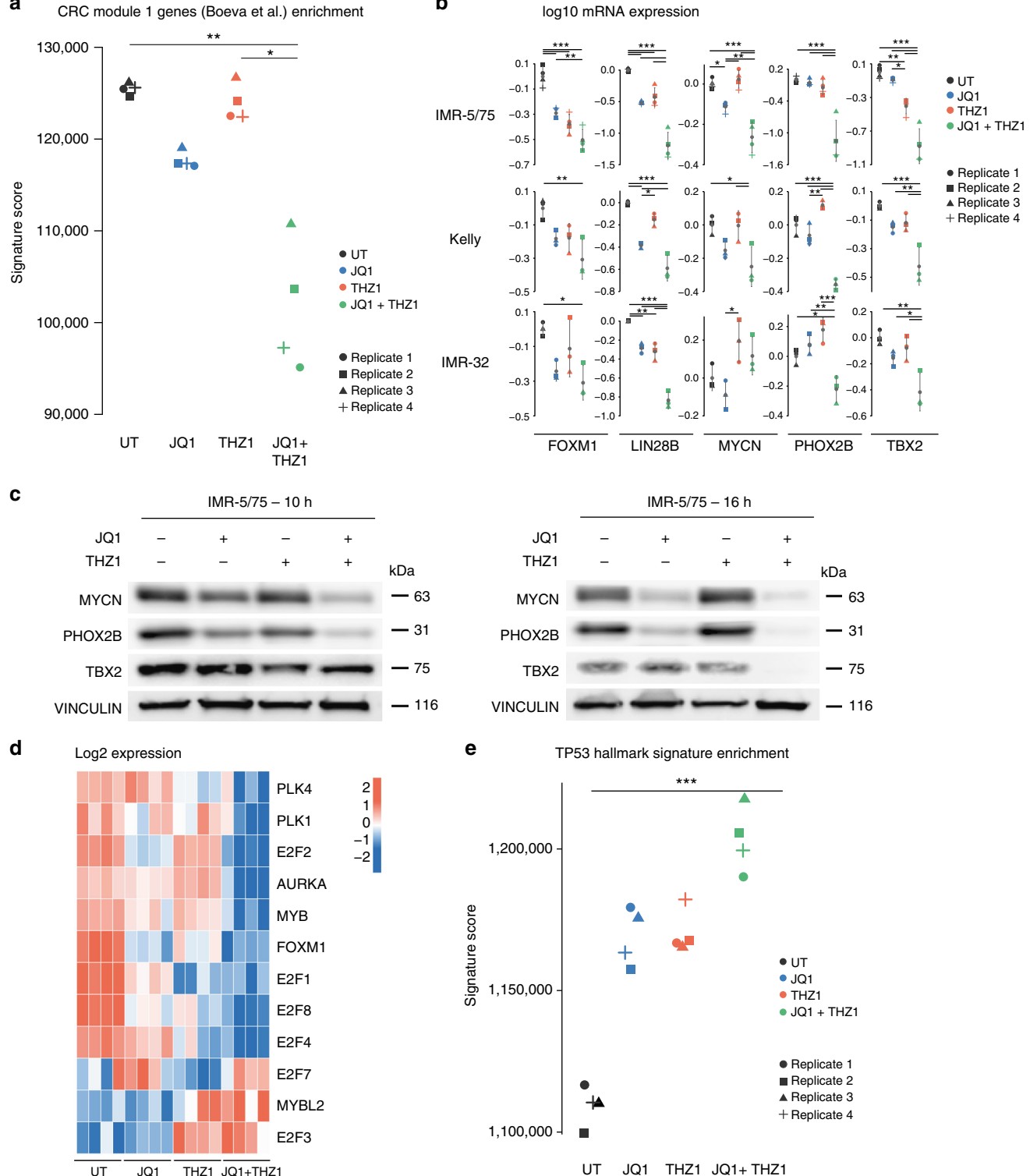

**Fig. 8** Downregulation of the CRC upon combined CDK7-BET inhibition. **a** Significant downregulation of the Module 1 CRC gene signature[15] upon treatment with 1 µM JQ1, 35 nM THZ1 and the combination in the IMR-5/75 cell line for 10 h. Kruskal–Wallis followed by a post-hoc Dunn's multiple comparisons test (four biological replicates per condition). **b** log10 *TBX2, FOXM1, PHOX2B, MYCN,* and *LIN28B* mRNA levels 10 h upon treatment with 1 µM JQ1, 35 nM THZ1 and the combination of JQ1 and THZ1 in the IMR-5/75, Kelly and IMR-32 cell lines. Error bars represent the s.d. of the three or four biological replicates. ANOVA statistical analysis followed by a post-hoc Tukey's test for multiple comparisons. **c** MYCN, PHOX2B, and TBX2 protein levels 10 h and 16 h upon treatment with 1 µM JQ1, 35 nM THZ1 and the combination of JQ1 and THZ1 in the IMR-5/75 cell line. **d** Heatmap showing the expression levels for the E2F-Dream complex core members upon treatment with 1 µM JQ1, 35 nM THZ1 and the combination of JQ1 and THZ1 in the IMR-5/75 cell line. Kruskal–Wallis followed by a post-hoc Dunn's multiple comparisons test (four biological replicates per condition). **e** Significant upregulation of the TP53 Hallmark geneset (MsigDB) upon treatment with JQ1, THZ1 and the combination in the IMR-5/75 cell line for 10 h. Kruskal–Wallis followed by a post-hoc Dunn's multiple comparisons test (four biological replicates per condition). *$p < 0.05$, **$p < 0.01$, ***$p < 0.001$

the CRC to the complex NB phenotype is limited, some insights are emerging. First, the work by Zeid et al.[16] revealed MYCN invasion of enhancers with canonical and non-canonical MYC (N) recognition sites in a dose-dependent fashion as one major driving event for the oncogenic CRC[16]. Secondly, GATA3 has been shown to act in de novo enhancer establishment as a pioneering transcription factor[51]. Thirdly, *PHOX2B* and *HAND2* are typical lineage identity genes[52] for which it can be assumed MYCN hyperactivity perturbs their normal epigenetic dynamic regulation leading to differentiation arrest. Here, we add *TBX2* as a novel actor in this complex circuitry. Although TBX2 has some properties of a lineage dependency factor, overexpression and amplification of *TBX2* has been associated with poor prognosis in various other cancer entities and *TBX2* is also a well-known developmental gene acting in various tissues including neural crest[53,54]. TBX2 has been described as a potent growth-promoting factor, partly due to its ability to bypass senescence and to repress key negative regulators of the cell cycle such as p21[20], which was confirmed as a repressed TBX2 target gene in the present study.

In keeping with this finding, TBX2 repressed genes in both CLB-GA and IMR-5/75 NB cells were found to be enriched for TP53 targets while common activated genes were enriched for FOXM1 targets, more specifically p53–p21–DREAM–E2F/CHR pathway known to control G2/M transition[55]. A number of the FOXM1 G2/M target genes are of particular interest, *i.e.* BRIP1 and *RRM2* which are implicated in control of replicative stress in NB (unpublished data), RAD51 implicated in homologous DNA repair and FANCI, FANCD2 and FANCG involved in replication fork stability and interstrand cross-link DNA repair. Together with data from previous reports, we propose that TBX2 repression of p21 enforces MYCN controlled p21 repression[56,58] which is followed by activation of CDK2/cyclin A, p107/p130 phosphorylation and finally DREAM complex repression. More directly, both MYCN and TBX2 activate FOXM1 expression and activity. In summary, we speculate that TBX2 plays a crucial role within the NB CRC, in a cooperative MYCN/TBX2 regulated p53-p21-DREAM-CDE/CHR pathway controlling G2/M cell cycle genes.Together with data from previous reports, we propose that TBX2 repression of p21 enforces MYCN controlled p21 repression[56] which is followed by activation of CDK2/cyclin A, p107/p130 phosphorylation and finally DREAM complex repression. More directly, both MYCN and TBX2 activate *FOXM1* expression and activity. In summary, we speculate that *TBX2* plays a crucial role within the NB CRC, in a cooperative *MYCN/TBX2* regulated p53-p21-DREAM-CDE/CHR pathway controlling G2/M cell cycle genes[57–59]. Finally, a function in epithelial-mesenchymal transition (EMT) has been attributed to TBX2[22]. While we found no evidence for a direct role for TBX2 in transition from adrenergic towards mesenchymal/neural crest-like phenotype of NB cells, in NB tumors *TBX2* expression levels were highly negatively correlated with those for *ZEB2*, known to be implicated in cell fate switch and EMT[60].

In this study, we also identified *TBX2-AS1* as a long noncoding RNA tightly co-expressed with *TBX2* suggesting cis-regulation, but its expression is unexpectedly increased upon *TBX2* and *MYCN* KD, possibly due to a compensation mechanism in response of attenuated *TBX2* and *MYCN* expression levels. Of further notice, this lncRNA has been recently proposed as a highly conserved tapRNA. Interestingly, other CRC genes such as *PHOX2B*, *GATA3*, and *HAND2* are also marked by nearby bidirectionally transcribed putative tapRNAs. Whether this is a common component of the complex regulation of CRC constituents remains to be studied.

Transcriptional addiction is emerging as an important novel drug vulnerability in cancer. Targeting multiple CRC constituents

and SE driven downstream genes may prevent resistance or enhance sensitivity to BRD4 inhibitors, as JQ1 blocks the binding of BRD4 with acetylated histones, rather than targeting BRD4 itself, which is still recruited to SEs and contributing to phase separation of chromatin domains[50]. Based on this rationale, we tested transcription addiction as a possible "Achilles heel" for the highly active transcribed genes in NB[61] and observed strong synergistic effects on cell growth and apoptosis for combined BET and CDK7 inhibition. By combining JQ1 and THZ1, we were able to target and induce a collapse of the SE-driven transcriptome including *TBX2* and *MYCN*. Another potential mechanism explaining the observed synergism could be modulating the p53 transcriptional program by JQ1 or *TBX2* inhibition, which sensitize the cells for CDK7 inhibition, as described recently in colon cancer[44], rather than targeting a dependency on transcriptional activation by TBX2 itself. In accordance with the strong *TBX2* downregulation, JQ1/THZ1 combination drugging affected the FOXM1-DREAM regulated target genes. Finally, several additional downregulated genes could also be envisioned to contribute to the observed drug synergism. The chromatin remodeler BPTF is a MYC interactor required for MYC chromatin recruitment and transcriptional activity[62]. The chromatin remodeler IWS1 is the binding partner of SPT6, an interaction which is induced by P-TEFb phosphorylation of RNAPII CTD and results in binding of the ALYREF mRNA export adaptor through binding nascent mRNA[63]. ID1 is a member of helix-loop-helix (HLH) family of proteins that regulate gene transcription through inhibitory binding to basic-HLH transcription factors and is involved in the repression of cell differentiation and activation of cell growth[64]. INO80 is an ATP-dependent chromatin remodeling complex involved in transcriptional regulation, DNA repair, replication fork stabilization, and restart[57]. In the list of upregulated genes one of the most differentially regulated ones is *HEXIM1*. This is intriguing given the well-established role of HEXIM1 in the switch of transcriptional programs through inhibiting P-TEFb and as sensitizer for BET inhibition. In addition, overexpression of *HEXIM1* upregulates expression levels of *p53* and p53 target genes by blocking p53 ubiquitination mediated by MDM2.

Of further notice, the observed drug synergism was particularly strong in primary tumor-derived cell lines (organoids) established prior to therapy. This further supports the importance for expanding in vitro testing with cell lines closer resembling the real clinical situation for optimal assessment of drugging effects.

In conclusion, we identified *TBX2* as a novel CRC constituent in high-risk NB contributing to the proliferative cellular state of these cells and offering opportunities for novel drugging strategies. Our data pave the way for more in-depth studies towards understanding the effects of drugging of transcriptional addiction as a guide towards novel therapies for NB.

## Methods

**Cell culture and generation of stable cell lines**. All patient specimens and samples were used in accordance with institutional and national policies, with appropriate approval provided by the relevant ethical committees at the respective institutions. All patient-related information was anonymized. All NB cell lines used in this manuscript (genotype and mutation status in Supplementary Table 1), the HEK-293TN and MCF-7 cell line were grown in RPMI1640 medium supplemented with 10% fetal bovine serum (FBS), 2 mM L-Glutamine and 100 IU/ml penicillin/streptavidin (referred further as complete medium) at 37 °C in a 5% $CO_2$ humid atmosphere. The IMR-5/75 shMYCN cell line was grown as previously described[58] with Tetracyclin-free FBS to avoid leakage in experiments where KD of MYCN was not desired; as such when we refer to IMR-5/75 in some experiments, this means that the IMR-5/75 shMYCN cells were grown in Tetracyclin-free medium. Short tandem repeat (STR) genotyping was used to validate cell line authenticity prior to performing the described experiments and Mycoplasma testing was done on a monthly basis. Patient-derived neuroblastoma tumor organoids (Kholosy et al., unpublished) were grown in Dulbecco's modified Eagle's medium (DMEM)-GlutaMAX containing low glucose and supplemented with 20% (v/v) Ham's F-12 Nutrient Mixture, B-27 Supplement minus vitamin A, N-2

Supplement, 100 IU/ml penicillin, 100 μg/ml streptomycin, 20 ng/mL epidermal growth factor (EGF), 40 ng/ml fibroblast growth factor-basic (FGF-2), 200 ng/ml insulin-like growth factor-1 (IGF-1), 10 ng/ml platelet-derived growth factor-AA (PDGF-AA) and 10 ng/ml platelet-derived growth factor-BB (PDGF-BB). EGF, FGF-2, PDGF-AA, and PDGF-BB were obtained from PeproTech and IGF-1 was obtained from R&D Systems. Other cell culture related materials were obtained from Life Technologies.

Four different mission shRNAs from the TRC1 library (Sigma-Aldrich, TRCN0000014824, TRCN0000014825, TRCN0000014826, TRCN0000014827, referred in the manuscript as sh1, sh2, sh3, sh4 respectively) targeting TBX2 and one non-targeting shRNA control (SHC002, NTC) were used to generate NB cell lines with TBX2 KD. Virus was produced by seeding $3 \times 10^6$ HEK-293TN cells in a $10 \text{ cm}^2$ dish 24 h prior to transfection. Transfection of the cells was done with trans-lentiviral packaging mix and lentiviral transfection vector DNA according to the Trans-Lentiviral shRNA Packaging Kit (Dharmacon) using CaCl$_2$ and 2x HBSS. 16 h after transfection cells were refreshed with reduced serum DMEM medium containing 5% FBS and the lentivirus-containing medium was harvested 48 h later. Virus was concentrated by adding 2500 μl ice-cold PEG-IT (System Biosciences) to 10 ml harvested viral supernatants, this was incubated overnight at 4 °C and complete medium was added to the remaining pellet upon centrifugation. IMR-32, CLB-GA, and SK-N-AS cells were transduced with concentrated virus (4 shRNAs and NTC) and 24 h after transduction cells were refreshed with medium and 48 h after transduction, cells were selected using 0.5–1 μg/ml puromycin. Transcriptomic and phenotypic read-outs were performed 7 days upon transduction. IMR-5/75 shMYCN cell lines were transduced with concentrated virus (4 shRNAs and NTC) and 24 h after transduction cells were refreshed with medium either with 1 ug/ml doxycycline or not. Puromycin selection was started 48 h upon transduction. 72 h upon transduction and 48 h upon shMYCN induction with doxycycline, cells were evaluated for transcriptomic and phenotypic changes.

**Real-time quantitative PCR**. Total RNA was extracted using miRNeasy kit (Qiagen) according to the manufacturer's instructions, including on-column DNase treatment, and concentration was determined with the Nanodrop (Thermo Scientific). cDNA synthesis was performed using the iScript Advanced cDNA synthesis kit from BioRad. PCR mix contained 5 ng of cDNA, 2.5 ul SsoAdvanced SYBR qPCR supermix (Bio-Rad) and 0.25 μl forward and reverse primer (to a final concentration of 250 nM, IDT) and was analysed on the LC-480 device (Roche) for RT-qPCR cycling. Expression levels were normalized using expression data of 3 stable reference genes out of 5 reference genes tested (SDHA, YWHAZ, TBP, B2M, HPRT1) and analyzed using qBasePlus software (http://www.biogazelle.com). All primer pair sequences can be found in Supplementary Table 2.

**Analysis of RNA-sequencing data**. RNA quality was determined with the Experion automated electrophoresis system (BioRad) prior to profiling. 250 ng of RNA isolated from cell line CLB-GA upon KD of TBX2 (three biological replicates per condition) and from cell line IMR-5/75 upon KD of TBX2 and/or MYCN (six biological replicates per condition) was used as input for library preparation with the TruSeq Stranded mRNA Sample Prep Kit from Illumina. 100 ng of RNA from the THZ1, JQ1 and combined drugging was used to perform an Illumina sequencing library preparation using the QuantSeq 3′ mRNA-Seq Library Prep Kit (Lexogen, Vienna, Austria) according to manufacturer's protocol. During library preparation 15 PCR cycles were used. Size distribution and quality was evaluated with a high sensitivity DNA ChIP on the bio-analyzer (Agilent) and qPCR quantification of the libraries using the Illumina Kapa Library quantification kit (Lightcycler 480 qPCR mix Kapa). RNA-seq libraries were sequenced on the NextSeq 500 platform (Illumina) using the Nextseq 500 High Output kit V2 75 cycles single-end (Illumina). Sample and read quality was checked with FastQC (v0.11.3). The QuantSeq generated reads were trimmed using cutadapt version 1.11 to remove the "QuantSEQ FWD" adaptor sequence. Reads were subsequently aligned to the human genome GRCh38 with STAR aligner (v2.5.2b and v2.5.3a). Final gene count values were obtained with RSEM (v1.2.31), which takes read mapping uncertainty into account. Non-locus strand specific read counts were filtered. To explore if the samples from different groups clustered together and to detect outlier samples, Principal Component Analyses (PCAs) on rlog transformed counts were performed using the R statistical computing software. Genes were only retained if they were expressed at counts per million (cpm) above one in at least four, eight or twelve samples for the drugging, shTBX2 in IMR-5/75 and shTBX2 in CLB-GA datasets respectively. Counts were normalized with the TMM method (R-package edgeR), followed by voom transformation and differential expression analysis using limma (R-package limma). A general linear model was built with the treatment groups (drugging or KD) and the replicates as a batch effect. Statistical testing was done using the empirical Bayes quasi-likelihood F-test. GSEA[59] was performed on the genes ordered according to differential expression statistic value (t). Signature scores were conducted using a rank-scoring algorithm[65]. Limma voom barcodeplots were used for visualization of gene set enrichment. In order to identify enriched functional classes and (upstream) co-regulators, iRegulon[24] and enrichR[66] were used with the default settings.

**Western blot analysis**. Proteins were isolated using a RIPA lysis buffer (5 mg/ml sodium deoxycholate, 150 mM NaCl, 50 mM Tris-HCl pH 7.5, 0,01% SDS solution, 0,1% NP-40) supplemented with protease inhibitors. In total, 40 μg of protein lysate was loaded onto an SDS-PAGE gel (10% Pre-cast, Bio-Rad), run for 1 h at 150 V and subsequently blotted onto a nitrocellulose membrane. The membranes were probed with the following primary antibodies: anti-TBX2 antibody (SC-17880, Santa Cruz, 1:1000 dilution / sc-514291, Santa Cruz, 1:1000 dilution), anti-PHOX2B antibody (sc-376997, Santa Cruz, 1:500 dilution), anti-MYCN antibody (SC-53993, Santa Cruz, 1:1000 dilution), anti-FOXM1 antibody (5436 S, Cell Signalling, 1:1000 dilution). As secondary antibody, we used HRP-labeled anti-rabbit (7074 S, Cell Signalling, 1:10,000 dilution) and anti-mouse (7076P2, Cell Signalling, 1:10,000 dilution) antibodies. Antibodies against Vinculin (V9131; Sigma-Aldrich, 1:10,000 dilution), alpha-Tubulin (T5168, Sigma-Aldrich, 1:10,000 dilution) or bèta actine (A2228; Sigma-Aldrich, 1:10,000 dilution) were used as loading control. All antibodies were diluted in milk/TBST (5 % non-fat dry milk in TBS with 0.1 % Tween-20). Binding of the antibodies with the membrane was evaluated using the SuperSignal West Dura Extended Duration Substrate (Thermo Scientific). Pictures were taken with the ChemiDoc-It Imaging System (UVP) using the VisionWorks analysis software (UVP), quantification of the blots were performed using ImageJ. Uncropped scans of the blots used in the main figures can be found in the Supplementary Figure 9.

**Phenotypic assessment of cells**. For colony formation assays, 2000 viable CLB-GA, IMR-32 and SK-N-AS cells with or without TBX2 KD were seeded in a 6-cm dish in a total volume of 5 ml complete medium and were then left unaffected for 10–14 days in a humid incubator at 37 °C. After an initial evaluation under the microscope, the colonies were stained with 0.005% crystal violet and digitally counted using ImageJ. The IncuCyte® Live Cell imaging system (Essen BioScience) was used for assessment of proliferation after TBX2 KD. Briefly, $17.5 \times 10^3$ viable CLB-GA and $15 \times 10^3$ IMR-32, SK-N-AS and IMR-5/75 cells, with or without TBX2 KD, were seeded in five replicates in a 96-well plate (Corning costar 3596) containing complete medium. Cell viability was measured in real-time using the IncuCyte by taking photos every 2 h of the whole well ( × 4). Masking was done using the IncuCyte® ZOOM Software. For cell cycle analysis, $7 \times 10^5$ cells were seeded in a T25 in complete medium and transduced with TBX2 shRNAs and controls and selected with puromycin, as described above. Cells were trypsinized and washed with PBS. The cells were resuspended in 300 μl cold PBS and while vortexing, 700 μl of 70% ice-cold ethanol was added dropwise to fix the cells. Following incubation of the sample for minimum 1 h at −20 °C, cells were washed in PBS and resuspended in 500 μl PBS with RNase A to a final concentration of 0.25 mg/ml. Upon 1 h incubation at 37 °C, 20 μl Propidium Iodide solution was added to a final concentration of 40 μg/ml. Samples were loaded on a BioRad S3$^\text{TM}$ Cell sorter and analysed with the Dean-Jett-Fox algorithm for cell-cycle analysis using the FlowJo® software package.

**Monitoring of synergistic effects of drug combinations**. The combined effect of JQ1 (MedChem Express) and THZ1 hydrochloride salt (Medchem express) in classical NB cell lines and patient-derived NB organoids was determined in a checkerboard fashion. Cell lines and organoids were seeded in 384-well plates and incubated overnight. Next, cell lines and organoids were co-treated with three-fold dilution series of JQ1 (5.1 nM–33.3 μM) and THZ1 (0.051 nM–0.333 μM), using the HP D300 Digital Dispenser (Tecan). Control cells were treated with DMSO. Cell viability was measured after 72 h (classical cell lines) or 120 h (organoids) using the 3-(4.5-dimethylthiazol-2-yl)-2,5-diphenyltetrazolium bromide (MTT) colorimetric assay (classical cell lines) or the CellTiter-Glo® luminescent assay from Promega (organoids). Cell viabilities of DMSO-treated cells were set to 100%. Excess over Bliss scores (score >0 indicates drug synergy, <0 indicates drug antagonism) were calculated and subsequently visualized using the R package synergyfinder. Values more than 100% were replaced by 100% for the concentration range experiments of SH-SY5Y and SK-N-BE(2c) to fulfill the criteria for Bliss score calculation.

**IncuCyte® assay for assessment of drug synergism**. Neuroblastoma cells (CLB-GA, IMR-32, IMR-5/75, SH-SY5Y, Kelly, SK-N-BE(2)-C and SK-N-AS) and MCF-7 cells were seeded in complete medium and a 1:200 dilution of IncuCyte® Annexin V Red Reagent for apoptosis (Essen Bioscience) in 96-well tissue culture plates (Corning costar 3596) in triplicate at 30% confluency and allowed to recover overnight. The next day, cells were treated with the indicated concentrations of JQ1 (BPS Bioscience), THZ1 hydrochloride salt (Medchem express) or Panobinostat (LBH589, Selleck Chemicals), or combinations thereof. Cell proliferation and cell death (AnnexinV positivity) were assessed continuously for 68–96 h after treatment by using the IncuCyte® Live Cell imaging system (Essen BioScience), by taking photos every 3–4 h of the whole well. Masking was done using the IncuCyte® ZOOM Software. Possible synergism was calculated using the Bliss method.

**Chromatine immunoprecipitation (ChIP) assay**. A total of $3–5 \times 10^7$ cells were crosslinked in complete medium with 1% formaldehyde while shaking for 7 min at room temperature. Crosslinking was quenched with 125 mM glycine (Sigma-Aldrich), cells were washed twice with PBS and stored at −80 °C. Cells were lysed

and sonicated with the M220 Covaris for 15 min to obtain 200–300 bp long fragments. Chromatin fragments were immunoprecipitated overnight using 1 μg for every $10^7$ cells of the following antibodies anti-TBX2 (SC-17880), anti-H3K4Me3 (Ab8580), anti-H3K4me1 (Ab8895), anti-H3K27ac (Ab4729) and anti-MYCN (SC-53993). 20 μl Protein A UltraLink® Resin beads was added per $10^7$ cells. Beads were eluted and split into two subsamples, one was used for western blot after adding 2x laemmli buffer and incubation at 95 °C for 10 min at 1000 rpm to elute the proteins from the beads, while the other subset was used for DNA isolation after resuspension in elution buffer and incubation for 30 min at 65 °C while vortexing every 2 min. Reverse crosslinking was done at 65 °C for 15 h. The chromatin for DNA purposes was resuspended in TE-buffer to dilute SDS in the elution buffer, incubated for 2 h at 37 °C with 0.2 mg/ml RNase and followed by an incubation of 2 h at 55 °C with 0.2 mg/ml proteinase K. DNA was isolated using 400 μl phenol:chloroform:isoamylalcohol (P:C:IA) in phase lock gel tubes (5Prime). Upon centrifugation, the aqueous layer was transferred to a new tube with 200 mM NaCl, 30 μg glycogen and 800 μl 100% ethanol, and incubated for 30 min at −20 °C. Upon centrifugation, the pellet was washed with 80% Ethanol and resuspended in RNase/DNase free water. Concentration was measured using the Qubit® dsDNA HS Assay Kit (Thermo Scientific). Library prep was done using the NEBNext Ultra DNA library Prep Kit from Illumina with 50–500 ng starting material and using 8–12 PCR cycles according to the manufacturer's instructions. Libraries were evaluated on the bio-analyzer using the Agilent high-sensitivity kit, followed by Pippin Prep™ (Sopachem) with a 2% Dye Free Marker L Agarose Gel Cassette to remove large fragments (>600 bp) or adapter dimers (<200 bp). Library concentrations were measured with the Illumina Kapa Library quantification kit (Lightcycler 480 qPCR mix) and ChIP-seq libraries were sequenced on the NextSeq 500 or HiSeq 2000 platform (Illumina) using the Nextseq 500 High Output kit V2 75 cycles single-end (Illumina) or HiSeq SBS Kit V4 50 cycles.

**Assay for transposase-accessible chromatin using sequencing**. ATAC-seq (assay for transposase-accessible chromatin using sequencing) was performed as previously described with minor changes[67]. In short, 50,000 cells were lysed and fragmented using digitonin and Tn5 transposase (Illumina). Next, the samples were purified using the MinElute kit (Qiagen). The transposed DNA fragments were amplified and purified using Agencourt AMPure XP beads (Beckamn Coulter). Library concentrations were measured with the Illumina Kapa Library quantification kit (Lightcycler 480 qPCR mix) and ATAC-seq libraries were sequenced on the NextSeq 500 platform (Illumina) using the Nextseq 500 High Output kit V2 75 cycles single-end (Illumina).

**ChIP-seq and ATAC-seq data-processing and analysis**. Prior to mapping to the human reference genome (hg19) with bowtie2 (v.2.3.1), quality of the raw sequencing data was evaluated using FastQC and adapter dimers were removed using cutadapt when necessary. Peak calling was performed using MACS2 taking a $q$ value of 0.05 as threshold. IGV was used for visualization purposes with the by IGVtools generated tdf files. Bedgraph files generated by MACS2 were converted to bigwig files and used as input for the ChIP-tracks in the figures, further processed using the R package Sushiplot. Homer[68] was used to perform motif enrichment analysis, taking 200 bp around the peak summit as input. Gene annotation tables, tag density plots, heatmaps and overlap of peaks maps were generated using Homer and the R packages EnrichedHeatmap and ChIPpeakAnno, with default parameters.

**Super-enhancer analysis**. ChIP-sequencing data of three different datasets were used for SE calling. (a) H3K27ac ChIP-sequencing for the cell lines CLB-GA (rep1), NGP, IMR-5/75 and IMR-32 (rep1) was performed as reported above with the H3K27ac antibody (ab4729, Abcam). (b) H3K27ac ChIP-sequencing for the cell lines SK-N-AS (rep1), CHP-212 (rep1), GI-M-EN (rep1) and IMR-5 was performed by Zymo Research using between $5 \times 10^6$ and $10 \times 10^6$ frozen cross-linked cells and pull down with the anti-H3K27ac antibody, as previously reported by Henssen et al[69]. (c) The public available H3K27ac ChIP-seq fastq files for CHP-212, CLB-MA, CLB-PE, CLB-GA (rep2 and rep3), GI-C-AN, GI-M-EN (rep2), IMR-32 (rep2), LA-N-1, N206, NB-EBc1, SH-SY5Y, SJNB-1, SJNB-12, SJNB-6, SJNB-8, SK-N-AS (rep2), SK-N-BE(2)-C, SK-N-DZ, SK-N-FI, TR14 (GSE90683[15]) and MCF-7 (GSE69112[70]) were obtained using the SRAtoolkit and mapping to ref genome hg19 was done using bowtie2 (v.2.3.1) with default settings. All BAM sequence files were mapped to the reference genome hg19 and subsequently downsampled using the picard tool DownsampleSam to obtain approximately 30 million reads per sample in order to be able to compare the H3K27ac activity between the cell lines. Cell line replicates are clustering together according to H3K27ac signal (Supplementary Fig. 1a), represented by a correlation (Pearson) heatmap generated by DeepTools.

SEs were defined with the Lilly algorithm[15] with the LILY provided configuration file for single-end reads and a merge distance of 200 bp. ControlFREEC was run with a window size of 50,000 and ploidy constraint of 2–4. All SEs that are overlapping across the 26 NB cell lines or that are within maximum distance of 500 bp from each other were clustered using bedtools, resulting in 176 SE clusters on chr17q (38.1 Mb—qter). Among the top 500 SEs ranked according to the median rank, six regions were prioritized based on the presence of an SE

cluster in min 20 NB cell lines. Ensembl genes (bioMart) in a 500 Kb range around the SE cluster were annotated to the different SE clusters. Of the annotated transcription factors[71,72], only those with a H3K27ac activity called peak at their TSS (Macs2, $q < 0.05$) were selected, as SEs are supposed to highly activate their target genes[13]. Genes were only selected when they had a H3K27ac marked TSS in all cell lines harboring the SE. Both IGV and the R package Sushi were used to visualize the SE regions in the different cell lines.

**4C-sequencing and data analysis**. Preparation of 4C samples was performed as described previously[73], starting with $10^7$ separated single cells that were cross-linked with 2% formaldehyde. DNA was digested using DpnII as a primary and Csp6I as a secondary restriction enzyme. 50 ng of the final 4C template of two NB cell lines (CLB-GA and SK-N-AS) was used for the amplification step with three different primer sets (Supplementary Table 2) designed for the TBX2 locus. PCR products were purified and pooled for multiplexed sequencing after adding 40% PhiX to increase sequence complexity. Sequencing was performed on the HiSeq 2000 V4 platform (50 bps single reads). Sequencing reads from the 4C library were demultiplexed using the demultiplex.py script (https://gist.github.com/meren/7632184) and aligned using bowtie (v.2.3.1) with parameters—sensitive—time—end-to-end -q -p 2. Aligned data were stored together with the metadata into a FourCSeq object[74]. The aligned libraries were subjected to a QC according to criteria published elsewhere[73]. Cis- and trans-interactions were determined using a custom script implementing the method described in[73,74]. Briefly, a $z$-score is computed on the binarized signal (i.e., counting all fragments covered by reads) using sliding windows (w = 100 fragments, W = 3000 fragments), and an FDR is computed from this $z$-score to identify significant interactions. We use a FDR of 1% for cis-interactions and 0.5% for trans-interactions. Domainograms showing the signal intensity of different ranges are plotted using a custom R-script. Bigwig files for visualization are generated using a smoothing over 21 consecutive fragments, and normalized to the total library size.

**TBX2 expression analysis in NB tumors and cell lines**. TBX2 expression analysis was performed on 283 NB tumors for which copy number ($n = 218$), mRNA expression ($n = 283$) and patient survival ($n = 276$) data were available from the Neuroblastoma Research Consortium (NRC, GSE85047), which is a collaboration between several European NB research groups. Additionally, the NB dataset from Su et al. ($n = 489$, GSE45547) was used as validation cohort[25]. All statistical analyses were performed using R (version 3.3.0). Genes were ranked according to the correlation of their expression (Spearman correlation) with TBX2 expression levels. Next, pre-ranked GSEA was performed as previously described[59] with the MsigDB c5.bp.v4.0 geneset and clustering of enriched genesets (only when FDR < 0.01, and R > 4/R < −3) was performed using the tool EnrichmentMap in Cytoscape (version 3.2.1). TBX2 expression in normal tissue, tumor, and cell line datasets was visualized on the R2 genomics analysis and visualization platform (http://r2.amc.nl/).

**DNA copy number analysis**. The presence of focal gains encompassing the TBX2 locus was investigated in published copy number profiling data of 556 high-risk tumors[29]. TBX2 focal gains were identified as copy number segments overlapping with the TBX2 locus with log2 ratio > = 0.3 and a maximal size of 5 Mb. Visual inspection of the segmented profiles was done using Vivar[75] and R2 (http://r2.amc.nl/).

Low coverage whole genome sequencing was performed following Library construction using the NuGen kit (Ovation® Ultralow Library Systems V2). In brief, 100 ng was used to prepare a Library and 150 bp paired sequenced in a flow cell using the MiSeq V3, to achieve an average of approximately 2 ×. Copy number analysis was performed using the FREEC tool[76].

**Quantification and statistical analysis**. Statistical significance of differences between conditions for the functional analysis colony forming assay and cell cycle was determined by a non-parametric Mann–Whitney test using R package (version 3.3.0) upon mean-centering the datapoints. The ANOVA (analysis of variance) test is used to assess how much of the variability in the TBX2 expression levels can be explained by the patient stage or/and TBX2 copy number status, while the non-parametric test Kruskal–Wallis followed by a post-hoc Dunn's multiple comparisons test was used to determine differences in gene expression and signatures scores between 4 different groups with 4 biological replicates per condition. Statistical significance of overlap between conditions was determined by Fisher test using R package. The non-parametric Spearman or parametric Pearson test was used for correlation analysis depending on the homoscedasticity assumption (Pearson if the assumption is met). Kaplan–Meier analysis with log-rank statistics was used for survival analysis. All assumptions for statistical analysis (performed using R) are met and as such justified. For qPCR experiments, reference genes were excluded if Genorm M value was greater than five and/or Coefficient of Variation greater than two, according to the qBaseplus software. For all experiments, at least three reference genes were used for the normalization according to good qPCR practice.

The details of quantification and statistical methods used can be found in each figure legend.

**Code availability**. The code for super-enhancer analysis is available at https://github.com/LabSpeleman/LILYconfig/blob/master/config.200.SE.txt and https://github.com/LabSpeleman/LILYconfig/blob/master/config.FREEC.general.txt

## Data availability

The RNA-sequencing, ChIP-sequencing and ATA-sequencing datasets generated during this study were deposited in the ArrayExpress database at EMBL-EBI (www.ebi.ac.uk/arrayexpress) with accession numbers: E-MTAB-6570, E-MTAB-6562, E-MTAB-6568, E-MTAB-6567 and E-MTAB-7025. Data that supports the findings of this study are available from the Neuroblastoma Research Consortium (NRC, GSE85047), Su et al. [GSE45547][25] and Depuydt et al. [GSE103123][29].

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

## Acknowledgements

The authors would like to thank W. Van Loocke, L. Mus, G. Dewyn, E. Sanders, J. Van Laere, E. De Smet, S. Claeys and E. Janssens for technical assistance and Prof. Stein Aerts for providing the transcription factor list used in iRegulon and iCistarget. This research was supported by the following funding agencies: the Belgian Foundation against Cancer (project 2015-146) to F.S.P., Ghent University (BOF10/GOA/019 and BOF16/GOA/23) to F.S.P., the Belgian Program of Interuniversity Poles of Attraction (IUAP Phase VII—P7/03) to F.S.P., the Fund for Scientific Research Flanders (Research projects G053012N, G050712N, and G051516N to F.S.P., G021415N to K.D.P and F.S.P.), 'Kom op tegen Kanker' (Stand up to Cancer) the Flemish cancer society (Research grant to F.S.P.), the European Union H2020 (OPTIMIZE-NB GOD9415N and TRANSCAN-ON THE TRAC GOD8815N to F.S.P.), 'Kinderkankerfonds' (Research grant to F.S.P.), Olivia Fund to F.S.P. and Villa Joep to F.S.P. FWO grants were supporting the work of B.D.C., S.L.N., S.C.L., K.D.R., C.N.N., S.V.H., and C.V.N ; K.V.B. and K.D.N. are supported by a BOF grant. D.R.M. is supported by a IWT grant. A.G.H. is supported by the Deutsche Forschungsgemeinschaft (DFG, Germ and Research Foundation – 398299703).

## Author contributions

B.D., F.S., K.D., G.D., and C.V. contributed to the conceptualization of the article; B.D., K.D., W.V., M.G., G.D., CV, K.V., K.D, E.D., W.K., G.S., V.B., R.V., J.V., J.M., P.M., J.D. and S.L. contributed to the development and design of methodology; B.D., C.V., C.H., G.D., C.N. and E.D. performed computational and statistical analysis; B.D., F.D., G.D., C.V., K.V., D.R., M.G., P.D., C.N., J.D., E.D., B.K., A.H. and G.S. performed experiments; V.B., G.S., M.G., F.W., D.D., A.H., E.D. and J.S. provided material, data and analysis tools; C.V. and B.D. managed the maintenance of data; B.D., F.S. and K.D. did write the original draft; G.D., S.V., C.V., K.D., S.L. and A.H. contributed to manuscript review and editing; B.D., G.D., and M.G. contributed to data representation and visualization; F.S. and K.D. directed the project and were responsible for funding.

## Additional information

**Competing interests:** The authors declare no competing interests.

