## [Peer Review File · Nature Communications]

Reviewers' comments:

Reviewer #1 (Remarks to the Author):

In this manuscript, the authors propose a new member of the MYCN-mediated CRC in neuroblastoma, TBX2. They show compelling data in support of their hypothesis indicating that the TBX2 gene is super-enhancer marked and rarely amplified implicating it in human disease as well as ChIP data indicating that TBX2 is in the CRC. They further show that TBX2 may act with MYCN through control of the FOXM1/E2F gene network. Finally, they propose and perform in vitro testing of a therapeutic intervention to combine CDK7 inhibition with BRD4 and HDAC inhibitors. Over all this is a solid, well written study with extensive data to support the author's hypotheses. Weaknesses include the lack on in vivo testing for the proposed therapeutic intervention, PDX and/or transgenic neuroblastoma efficacy testing should be done if the plan is to translate this data into the clinic.

Reviewer #2 (Remarks to the Author):

This manuscript takes an integrated genomics approach to identify the transcription factor TBX2 as a putative component of the "core regulatory circuitry" (CRC) in high-risk neuroblastoma (NB). The criteria used to make this assignment are 1) the presence of TBX2 within the Chromosome 17q region frequently gained in these tumors; 2) elevated levels of TBX2 expression in NB relative to normal tissues and other tumors; 3) the proximity of superenhancers (SEs) to the TBX2 gene, and chromatin looping interactions detected between the TBX2 promoter region and regions harboring these SEs in NB cells; and 4) evidence that TBX2 knockdown selectively impairs proliferation and perturbs gene expression in NB cells, both independently and in conjunction with depletion of MYCN. The data in support of TBX2-dependency of NB cell proliferation are solid, and the implication of TBX2 in NB will be of interest to workers in the field. Future in-depth studies of TBX2 might illuminate gene expression signatures that help determine the course and response to therapy of this common pediatric tumor. The study sheds little light on the mechanism by which TBX2-directed transcription promotes NB cell growth and survival, however, and the results obtained with small-molecule inhibitors of CDK7, BRD4 and HDACs, which seek to connect TBX2 "addiction" with established transcriptional dependencies in NB, are preliminary and seemingly contradictory of previous, more thorough studies. This piece of the story would need to be further developed if it is to be included. My specific concerns are:

1. The authors report that NB cells with single-copy MYCN are more sensitive to the CDK7 inhibitor THZ1 than are those with MYCN amplification (p. 13, Fig. 7 and Supp. Fig. 7). This directly contradicts a major conclusion of Chipumuro et al. (ref. 47), who used a larger number of NB lines (and a range of THZ1 doses) to show that THZ1 selectively killed cells with MYCN amplification. MYC amplification was also associated with heightened THZ1-sensitivity in small-cell lung cancer (Christensen et al., *Cancer Cell* 26: 909-22, 2014). This discrepancy needs to be addressed. It is perhaps relevant that the authors base many of their conclusions on treatments with a single dose of THZ1 (35 nM), and seem to obtain variable results even in a single MYCN-amplified cell line, IMR5-75 (cf. THZ1 effects on relative proliferation in Fig. 7b and 7c).

2. There are errors and inaccuracies in describing functions of the transcriptional machinery. For example, it is incorrect to describe the use of either CDK7 or BET inhibitors as "epigenetic drug targeting" as they do in the abstract. It is also not quite right to imply that the functions of BET domain proteins and CDK7 are neatly divided between transcription initiation and elongation, respectively (p. 3, last sentence of Introduction). An important function ascribed to BRD4 (the major target of JQ1) is recruitment of an elongation factor, P-TEFb.

3. Throughout the manuscript, the authors' end-of-paragraph conclusions and summary statements get ahead of their actual results. For example, they call TBX2 a "dependency factor" on p. 5, prior to knockdown results that are shown in Fig. 4. The same critique applies to the description of TBX2 as "a dosage-sensitive transcription factor" on p. 7 (again prior to any manipulation of TBX2 levels). In a similar vein, the authors overuse (or misuse) the terms "driver"

in reference to TBX2 (see for example last sentence on p. 10), and “cooperativity” with respect to how TBX2 and MYCN work together (“cooperation” is better).

4. There are suggestions that TBX2 knockdown or HDAC1 inhibition is triggering a p53 response (e.g. the GSEA in Fig. 4a). THZ1 synergizes with either of these treatments to slow proliferation or induce apoptosis, and might be doing so by modulating the p53 transcriptional program, an effect of CDK7 inhibitors recently reported in colorectal and other cancer cell types (Kalan et al., Cell Reports 21: 467-81, 2017). A priori this seems equally as likely as the preferred (I think) explanation, i.e., that THZ1 is targeting a dependency on transcriptional activation by TBX2, similar to what has been reported in other studies on T-ALL (Kwiatkowski et al., Nature 511: 616-20, 2014), NB (ref. 47), SCLC (Christensen et al., 2014) and triple-negative breast cancer (Wang et al., Cell 163:174-86, 2015). Both potential mechanisms should be discussed and properly referenced.

5. A minor point: a literal reading of the sentence beginning, “Moreover, TBX2 is marked...” (pp. 4-5) would mean that hNCC and MCF-7 are NB cell lines.

6. The figure showing results of “ANOVA analysis” (Fig. 2d) needs additional explanation. This is not a routine type of analysis that a general audience can be expected to know.

7. On p. 8, second paragraph, the authors single out CDKN1A as a TBX2 target gene without saying if it is up- or down-regulated upon knockdown. (It can be inferred from later results that it goes up, but this should be specified here.)

8. p. 11, last sentence: Please define “TH-MYCN.”

9. p. 12, the second sentence reads: “we observed a stronger decrease in cell proliferation and G1-cell phase arrest” which is clearly not correct. I think “increased” or “exacerbated” needs to be inserted before “G1.” (Also, the “cell” should be deleted.)

10. Also on p. 12, the authors mention MYC among the DNA-binding motifs enriched in genes upregulated upon combined knockdown of TBX2 and MYCN. This is counterintuitive and seems like it should be discussed in greater detail than the oblique reference to “feedback loops” much later in the same paragraph.

11. The text refers to a pro-apoptotic effect of combining THZ1 with the HDAC1 inhibitor panobinostat with a call-out to Fig. 7b (p. 13), which only shows effects on cell proliferation (not death). Annexin V staining results suggestive of apoptosis are only shown for the THZ1/JQ1 combination, and only in Supplementary Fig. 7b.

Response to Referees

We are very pleased to have received a favorable and constructive review of our manuscript NCOMMS-18-06938 entitled “TBX2 is a neuroblastoma core regulatory circuitry component enhancing MYCN/FOXM1 reactivation of DREAM targets” and we are thankful for the detailed review of this manuscript. In response to the constructive feedback from reviewers, we have carefully conducted several additional experiments, added our new data and adjusted the manuscript where necessary. We hope these substantial improvements will allow acceptance of our manuscript for publication in Nature Communications.

Below, we address each of the reviewers’ comments in detail, indicating where corresponding changes to the text of the manuscript have been made (highlighted in the manuscript itself), as well as any necessary modifications to the figures. References in the rebuttal are listed at the end and if appropriate also included in the adapted manuscript.

Response to Reviewer #1

In this manuscript, the authors propose a new member of the MYCN-mediated CRC in neuroblastoma, TBX2. They show compelling data in support of their hypothesis indicating that the TBX2 gene is super-enhancer marked and rarely amplified implicating it in human disease as well as ChIP data indicating that TBX2 is in the CRC. They further show that TBX2 may act with MYCN through control of the FOXM1/E2F gene network. Finally, they propose and perform in vitro testing of a therapeutic intervention to combine CDK7 inhibition with BRD4 and HDAC inhibitors. Over all this is a solid, well written study with extensive data to support the author's hypotheses. Weaknesses include the lack on in vivo testing for the proposed therapeutic intervention, PDX and/or transgenic neuroblastoma efficacy testing should be done if the plan is to translate this data into the clinic.

We thank the reviewer for the positive appreciation of our work. We fully agree that this work should boost further experiments towards clinical translation. In a first step towards this goal, we have considerably expanded the *in vitro* pre-clinical testing including the use of primary patient derived tumor cell lines (grown in stem cell medium, see below). Currently, THZ1 and JQ1 are tool compounds and preferably *in vivo* testing should be conducted using novel generation CDK7 and BET inhibitors. We are definitely aiming to explore this in the near future through initial testing in our zebrafish models (maximum tolerable doses for individual and combined drugs) followed by tests on patient derived xenografts in mice in collaboration with the Memorial Sloan Kettering Cancer Center (dr. S. Roberts). However, we think the current substantial and highly novel data set (including extensive novel data added in this revision) on the *in vitro* pre-clinical part provides substantial novel insights into the molecular basis of the observed drug synergism which we consider highly relevant for the research community and thus should not be further delayed for publication.

Response to Reviewer #2

Summary: *This manuscript takes an integrated genomics approach to identify the transcription factor TBX2 as a putative component of the “core regulatory circuitry” (CRC) in high-risk neuroblastoma (NB). The criteria used to make this assignment are 1) the presence of TBX2 within the Chromosome 17q region frequently gained in these tumors; 2) elevated levels of TBX2 expression in NB relative to normal tissues and other tumors; 3) the proximity of superenhancers (SEs) to the TBX2 gene, and chromatin looping interactions detected between the TBX2 promoter region and regions harboring these SEs in NB cells; and 4) evidence that TBX2 knockdown selectively impairs proliferation and perturbs gene expression in NB cells, both independently and in conjunction with depletion of MYCN. The data in support of TBX2-dependency of NB cell proliferation are solid, and the implication of TBX2 in NB will be of interest to workers in the field. Future in-depth studies of TBX2 might illuminate gene expression signatures that help determine the course and response to therapy of this common pediatric tumor. The study sheds little light on the mechanism by which TBX2-directed transcription promotes NB cell growth and survival, however, and the results obtained with small-molecule inhibitors of CDK7, BRD4 and HDACs, which seek to connect TBX2 “addiction” with established transcriptional dependencies in NB, are preliminary and seemingly contradictory of previous, more thorough studies. This piece of the story would need to be further developed if it is to be included. My specific concerns are:*

Authors Response: We thank the reviewer for the positive and constructive comments. We agree that additional insights into the underlying molecular basis of the observed synergism would be an added value to our paper. Therefore, we further interrogated the mechanism of combined CDK7 and BRD4 inhibition. Furthermore, we also clarified each of the issues raised by the reviewer with text updates (strikethrough when removed, underlined when added) and experimental revisions. We believe that this has significantly improved the manuscript. A point-by-point response is included below.

1. The authors report that NB cells with single-copy MYCN are more sensitive to the CDK7 inhibitor THZ1 than are those with MYCN amplification (p. 13, Fig. 7 and Supp. Fig. 7). This directly contradicts a major conclusion of Chipumuro et al. (ref. 47), who used a larger number of NB lines (and a range of THZ1 doses) to show that THZ1 selectively killed cells with MYCN amplification. MYC amplification was also associated with heightened THZ1-sensitivity in small-cell lung cancer (Christensen et al., Cancer Cell 26: 909-22, 2014). This discrepancy needs to be addressed. It is perhaps relevant that the authors base many of their conclusions on treatments with a single dose of THZ1 (35 nM), and seem to obtain variable results even in a single MYCN-amplified cell line, IMR5-75 (cf. THZ1 effects on relative proliferation in Fig. 7b and 7c).

In order to address several comments of the reviewers concerning our initial drugging experiments, we decided to perform a much broader screening including additional cell lines over a wide range of concentrations for both drug individually and combined (yielding a total of 96 data points for each cell line). These data are included in the revised manuscript and all sections have been extended as described in detail below.

A first concern was related to our observation that MYCN non-amplified cells responded stronger to THZ1 which, as noted by reviewer, is in contrast to the report of Chipumuro et al¹. A possible explanation for the **conflicting data** could be related to the lower drug concentration used in our initial experiments to specifically search for synergism at concentrations that would offer a broader therapeutic window for clinical application. Indeed, Kwiatkowski et al² described that THZ1 triggers apoptosis in tumor cells with fixed dependencies on oncogenic transcription factors at low doses (more CDK7 selective) while high doses effects global gene expression and Christensen et al³ showed that low dose of THZ1 resulted in a lower log fold change of steady-state mRNA levels, and more gene-selective effects. These reports thus support the need to test the effects on viability over a range of concentrations. When taking into account the concentrations used in the literature and observed responses and those in our expanded data set (Fig. 8a), our data are compliant with those reported. Therefore, we decided to remove this text from the original manuscript on p. 13:

~~Our data also suggest that MYCN single copy cell lines seem to be particularly sensitive to THZ1 while MYCN amplified cell lines seems to be only sensitive to JQ1, however, when combined, cell viability decreases dramatically (Fig. 7c and Supplementary Fig. 7a,b).~~

As requested by the reviewer and explained above, we extended our experiment on the NB cell lines SK-N-AS, CLB-GA, SH-SY5Y, SK-N-BE(2C) and Kelly with two different **concentration ranges for each compound** JQ1 (5.1 nM-33.3 μ M) and THZ1 (0.051 nM-0.333 μ M) according to literature^{1,4}. The highest synergism Bliss score for the classical cell lines is obtained with the concentrations underneath:

- SK-N-AS: 33 μ M JQ1 and 37 nM THZ1
- CLB-GA: 11 μ M JQ1 and 37 nM THZ1
- SH-SY5Y: 33 μ M JQ1 and 37 nM THZ1
- IMR-32: 137 nM JQ1 and 4.12 nM THZ1
- SK-N-BE(2c): 3.7 μ M JQ1 and 111 nM THZ1
- Kelly: 411 nM JQ1 and 37 nM THZ1

The drug response observations and synergism (Excess over bliss) results are depicted in fig 7a and shown below, and we adapted the text accordingly.

Fig 7a. Heatmap with the percentage of cell viability for serial dilutions of JQ1 (5.1 nM-33.3 μ M) in combination with THZ1 (0.051 nM-0.333 μ M) in Kelly, SK-N-BE(2c), SK-N-AS, IMR-32, CLB-GA and SHS-Y5Y, 72h upon treatment; and respective 3D representation of Excess over Bliss scores. Excess over Bliss scores > 0 indicates drug synergy, < 0 indicates drug antagonism. Data points in the screen represent mean of two technical replicates.

As a further positive note, it can be appreciated that our original concentrations of 35 nM for THZ1 and 1 μ M for JQ1 were well chosen for the classical cell lines. For JQ1 we notice variability between the cell lines, which is in line with reports in literature⁴, but overall indicating that MYCN amplified cells are more sensitive for JQ1 than MYCN non-amplified cells. Of further importance, the fixed drug concentrations we used for the response over time analysis shows synergism in all cell lines when evaluating the concentration in the checkerboard experiments.

Finally, in our opinion the inclusion of two neuroblastoma organoids (one MYCN amplified, one MYCN non-amplified) further widens the scope of our findings away from the classical cell lines derived from heavily treated patients towards primary tumor derived cell lines which more closely resemble the biology and drug response in a pre-treatment clinical situation. In this respect, it is important to notice that these organoids are much more sensitive for the THZ1/JQ1 drug combination (fig 7b: MYCN amplified organoid: highest synergism with 137 nM JQ1 and 12.3 nM THZ1; MYCN non-amplified organoid: highest synergism with 45 nM JQ1 and 12.3 nM THZ1).

Fig 7b. Heatmap with the percentage of cell viability for serial dilutions of JQ1 (5.1 nM-33.3 μ M) in combination with THZ1 (0.051 nM-0.333 μ M) in two organoids (one MYCN amplified and one MYCN non-amplified), 5 days upon treatment; and respective 3D representation of Excess over Bliss scores. Excess over Bliss scores > 0 indicates drug synergy, < 0 indicates drug antagonism. Data points in the screen represent mean of two technical replicates.

As indicated in Material and Methods and Results section, organoids were evaluated for response 5 days after treatment (as compared to 72h for the classical cell lines) which could be a possible explanation for the observed difference. Future in vivo testing could clarify this. Moreover, we have repeatedly observed that organoids are more sensitive for classical cytostatics than cell lines.

2. There are errors and inaccuracies in describing functions of the transcriptional machinery. For example, it is incorrect to describe the use of either CDK7 or BET inhibitors as “epigenetic drug targeting” as they do in the abstract. It is also not quite right to imply that the functions of BET domain proteins and CDK7 are neatly divided between transcription initiation and elongation, respectively (p. 3, last sentence of Introduction). An important function ascribed to BRD4 (the major target of JQ1) is recruitment of an elongation factor, P-TEFb.

Authors Response: In line with this comment, we have adapted the text:

> we changed the abstract as follow:

“Chromosome 17q gains are almost invariably present in high-risk neuroblastoma cases. We performed an integrative epigenomics search for dosage-sensitive transcription factors on 17q marked by H3K27ac defined super-enhancers and identify TBX2 as top candidate gene. We show that TBX2 is a constituent of the recently established core regulatory circuitry in neuroblastoma with features of a cell identity transcription factor, driving proliferation through activation of p21-DREAM repressed FOXM1 target genes. Combined MYCN/TBX2 knockdown enforces cell growth arrest suggesting that TBX2 enhances MYCN sustained activation of FOXM1 targets. Targeting transcriptional addiction by combined CDK7 and BET bromodomain inhibition shows synergistic effects on cell viability with strong repressive effects on CRC gene expression and p53 pathway response as well as several genes implicated in transcriptional regulation. In conclusion, we provide insight into the role of the TBX2 CRC gene in transcriptional dependency of neuroblastoma cells warranting clinical trials using BET and CDK7 inhibitors”

> in the introduction on p.4:

“Finally, we demonstrate that combined pharmacological targeting of transcription initiation and elongation transcriptional addiction using a BET and CDK7 inhibitor respectively, yields synergistic effects on TBX2 downregulation leading to massive apoptosis. “

3. Throughout the manuscript, the authors’ end-of-paragraph conclusions and summary statements get ahead of their actual results. For example, they call TBX2 a “dependency factor” on p. 5, prior to knockdown results that are shown in Fig. 4. The same critique applies to the description of TBX2 as “a dosage-sensitive transcription factor” on p. 7 (again prior to any manipulation of TBX2 levels). In a similar vein, the authors overuse (or misuse) the terms “driver” in

reference to TBX2 (see for example last sentence on p. 10), and “cooperativity” with respect to how TBX2 and MYCN work together (“cooperation” is better).

Authors Response: We agree with the reviewer’s concern of the use of “dependency factor, cooperativity and driver” and changed the text accordingly:

> the text on p.5:

“Taken together, our data suggest a key possible important role for TBX2 as a ~~possible important novel dependency factor mastering transcriptional control during~~ hitherto unrecognized transcriptional regulator implicated in NB tumor development.

> Text p.11 and title p.9: *“In summary, we have shown that TBX2 is implicated in cell cycle and proliferation through regulating driving an E2F-FOXM1 driven cellular state.”*

> Text p.12: *“To further experimentally explore this presumed cooperation cooperativity between TBX2 and MYCN, we assessed the effects of TBX2 KD in the presence of high versus low MYCN levels. “*

> Text p.12: *“In summary, our data support the cooperation cooperativity of TBX2 and MYCN regulating the NB driven proliferative cellular state mediated by FOXM1.”*

We understand the concern of the reviewer about the possible wrong annotation of TBX2 as “dosage-sensitive” gene. However, this statement refers to the clear and significant effects on TBX2 transcription levels versus copy number status (Fig. 2d).

4. There are suggestions that TBX2 knockdown or HDAC1 inhibition is triggering a p53 response (e.g. the GSEA in Fig. 4a). THZ1 synergizes with either of these treatments to slow proliferation or induce apoptosis, and might be doing so by modulating the p53 transcriptional program, an effect of CDK7 inhibitors recently reported in colorectal and other cancer cell types (Kalan et al., Cell Reports 21: 467-81, 2017). A priori this seems equally as likely as the preferred (I think) explanation, i.e., that THZ1 is targeting a dependency on transcriptional activation by TBX2, similar to what has been reported in other studies on T-ALL (Kwiatkowski et al., Nature 511: 616-20, 2014), NB (ref. 47), SCLC (Christensen et al., 2014) and triple-negative breast cancer (Wang et al., Cell 163:174-86, 2015). Both potential mechanisms should be discussed and properly referenced.

We thank the reviewer for this critical point of view. We need to clarify that we didn’t perform gene expression profiling upon HDAC1 inhibition as might be mistakenly understood by the reviewer (1st sentence). Out of our data we cannot conclude if HDAC1 inhibition is triggering a p53 response, but in literature this is extensively described⁵.

With the data described above, the reviewer might have a strong argument that THZ1 synergizes with JQ1 treatment to slow proliferation or induce apoptosis by modulating the p53 transcriptional program.

In response to this comment, we further elucidated the effects of JQ1 and THZ1 treatment on the transcriptome, by performing a comprehensive **RNA-sequencing analysis** 10h upon 1 μ M JQ1 and 35 nM THZ1 treatment and the combination in the IMR-5/75 cell line. Interestingly, P53 pathway is indeed enriched upon JQ1 treatment and THZ1 treatment, AND synergistically affected in the combination treatment group (Fig 8e) indicating that combining JQ1 with THZ1 enforces the p53 response, possibly by downregulating MDM2 and upregulating CDKN1A, which results in synthetic lethality as recently described by Kalan et al⁶. The reviewer is right that we should describe both potential mechanisms in the discussion. To reflect on this, we added following sentence on page 19:

“Another potential mechanism explaining the observed synergism could be modulating the p53 transcriptional program by JQ1 or TBX2 inhibition, which sensitize the cells for CDK7 inhibition, as described recently in colon cancer⁴⁸, rather than targeting a dependency on transcriptional activation by TBX2 itself. In accordance with the strong TBX2 downregulation, JQ1/THZ1 combination drugging affected the FOXM1-DREAM regulated target genes.”

In addition, we added a new paragraph to the result section on p. 14-15 and added Fig 8 and Supplementary Fig 8 to the manuscript.

Fig 8: Combined BET and CDK7 inhibition as a novel therapeutic approach for neuroblastoma

(a). Significant downregulation of the Module 1 CRC gene signature (Boeva et al., Nature Genetics, 2017) upon treatment with 1 μ M JQ1, 35 nM THZ1 and the combination in the IMR-5/75 cell line for 10 h.
 (b). log₁₀ TBX2, FOXM1, PHOX2B, MYCN and LIN28B mRNA levels 10 h upon treatment with 1 μ M JQ1, 35 nM THZ1 and the combination of JQ1 and THZ1 in the Kelly, IMR-5/75 and IMR-32 cell lines. Error bars represent the s.d. of the 3 biological replicates.
 (c). MYCN, PHOX2B and TBX2 protein levels 10 h and 16 h upon treatment with 1 μ M JQ1, 35 nM THZ1 and the combination of JQ1 and THZ1 in the IMR-5/75 cell line.
 (d). Heatmap showing the expression levels for the E2F-DREAM complex core members upon treatment with 1 μ M JQ1, 35 nM THZ1 and the combination of JQ1 and THZ1 in the IMR-5/75 cell line.
 (e). Significant upregulation of the TP53 Hallmark geneset (MsigDB) upon treatment with JQ1, THZ1 and the combination in the IMR-5/75 cell line for 10 h. Kruskal-Wallis followed by a post-hoc Dunn's multiple comparisons test * $p < 0.05$, ** $p < 0.01$, *** $p < 0.001$

Supplementary Fig 8:
 Combined BET and CDK7 inhibition as a novel therapeutic approach for neuroblastoma

(a). Global gene expression levels upon treatment with 1 μ M JQ1, 35 nM THZ1, and the combination in the IMR-5/75 cell line for 10 h, for every comparison represented by the log fold changes of expression (Combination vs control, JQ1 vs control, THZ1 vs control). **(b).** Line plot showing the log fold changes normalized to the control for a set of CRC genes upon treatment with 1 μ M JQ1, 35 nM THZ1 and the combination in the IMR-5/75 cell line for 10 h. **(c).** Significant downregulation of the CRC geneset in Kelly, NGP and SK-N-BE(2c) (Zeid et al., Nature Genetics, 2018) and adrenergic geneset (Van Groningen et al., Nature Genetics, 2017) upon treatment with 1 μ M JQ1, 35 nM THZ1 and the combination in the IMR-5/75 cell line for 10 h. **(d).** TBX2, PHOX2B and MYCN protein levels 10 h upon treatment with 1 μ M JQ1, 35 nM THZ1 and the combination in the IMR-5/75, Kelly and IMR-32 cell line. **(e).** Line plot showing the log fold changes normalized to the control for the E2F-Dream complex core members upon treatment with 1 μ M JQ1, 35 nM THZ1 and the combination of JQ1 and THZ1 in the IMR-5/75 cell line for 10 h. **(f).** HEXIM1 log₂ mRNA expression upon treatment with 1 μ M JQ1, 35 nM THZ1 and the combination in the IMR-5/75 cell line for 10 h. Kruskal-Wallis followed by a post-hoc Dunn's multiple comparisons test: * $p < 0.05$, ** $p < 0.01$, *** $p < 0.001$

Furthermore, we added 2 or 3 extra biological replicates to our qPCR dataset from original fig 7d upon THZ1, JQ1 treatment or the combination for the Kelly, IMR-32 and IMR-5/75 cell lines, where we confirm synergistically effects on FOXM1, LINB28B, MYCN, PHOX2B and TBX2 mRNA levels in 3 NB cell lines. The four replicates of the latter cell line were used for RNA-sequencing. The data are depicted in fig 8b.

Moreover, we also extensively repeated the western blots upon JQ1, THZ1 and combination treatment in Kelly, IMR-32 and IMR-5/75 and replaced the western blots from Fig 7e and Supplementary Fig 7c, and moved them to Fig 8c and Supplementary Fig 8d.

5. A minor point: a literal reading of the sentence beginning, “Moreover, TBX2 is marked...” (pp. 4-5) would mean that hNCC and MCF-7 are NB cell lines.

Authors Response: We adapted the text on p.5 as follows:

“Moreover, TBX2 is marked by a SE in all investigated NB cell lines but not in the ~~except for~~ human neural crest line (hNCC) and the MCF-7 breast cancer cell line (Fig. 1b and Supplementary Fig. 1b).”

6. The figure showing results of “ANOVA analysis” (Fig. 2d) needs additional explanation. This is not a routine type of analysis that a general audience can be expected to know.

Authors Response: We have added the following text to the material and methods section on page 33:

“The ANOVA (analysis of variance) allows to assess how much of the variability in the TBX2 expression levels can be explained by the patient stage or/and TBX2 copy number status.”

7. On p. 8, second paragraph, the authors single out CDKN1A as a TBX2 target gene without saying if it is up- or down-regulated upon knockdown. (It can be inferred from later results that it goes up, but this should be specified here.)

Authors Response: To clarify this we changed the text on p.9 as follows:

“1055 and 1326 genes were differentially down and upregulated respectively (adj.p.val < 0.05, Supplementary Table 2), including the known TBX2 target upregulated gene CDKN1A which is a known target gene repressed by TBX2.”⁷

8. p. 11, last sentence: Please define “TH-MYCN.”

Authors Response: We have adapted the text on page 12 as follows:

“significant dynamical downregulation of the shTBX2 signature was noted during Tg(TH-MYCN) driven NB formation in a transgenic mice, at one, two and six weeks after birth”⁸.

9. p. 12, the second sentence reads: “we observed a stronger decrease in cell proliferation and G1-cell phase arrest” which is clearly not correct. I think “increased” or “exacerbated” needs to be inserted before “G1.” (Also, the “cell” should be deleted.)

Authors Response: We agree with the proposed change and adapted the text on p. 12 as follows:

“we observed a stronger decrease in cell proliferation and increased G1-cell phase arrest”

10. Also on p. 12, the authors mention MYC among the DNA-binding motifs enriched in genes upregulated upon combined knockdown of TBX2 and MYCN. This is counterintuitive and seems like it should be discussed in greater detail than the oblique reference to “feedback loops” much later in the same paragraph.

Authors Response:

We thank the reviewer for this critical remark. Iregulon is an algorithm for motif detection and track discovery⁹ in a predefined set of genes. Motif detection is done for nearly ten thousand candidate motifs (from databases such as TRANSFAC or Homer), while more than one thousand ChIP-seq tracks (from databases such as encode) are used for track discovery.

The enrichment of EP300, REST, NANOG, MYC, and CTBP2 transcription factors in the upregulated genes upon *TBX2* and *MYCN* knockdown is actually the enrichment of binding of EP300, REST, NANOG, MYC and CTBP2 based on public available ChIP-seq data, more specifically EP300 in SKNSH, REST in H1 neurons, and NANOG, MYC, and CTBP2 in H1 ESC cells. This indicates that for example genes bound by NANOG, MYC and CTBP2 in human embryonal stem cells, are upregulated upon *TBX2* and *MYCN* knockdown. It needs to be clarified that MYC itself is not expressed in IMR-5/75 and doesn't change in expression upon *TBX2* or/and *MYCN* knockdown nor upon JQ1 and/or THZ1 treatment, which makes it very unlikely that MYC does bind the upregulated genes upon *TBX2* and *MYCN* knockdown in this neuroblastoma cells. As it is difficult to make conclusions out of the enrichment of 1 specific ChIP-seq dataset, we decided to retain only the gene hubs when there is enrichment for 1 or more ChIP-seq tracks and/or motif enrichment of that particular transcription factor. Therefore, we can only retain the hubs P300 and NANOG in the dataset with upregulated genes and FOXM1, MYBL2 and E2F4 in the dataset with downregulated genes. TFDP1 is a known interaction partner of E2F members, and for that reason TFDP1 is still a worth noting gene hub in the network of downregulated genes. As the network with upregulated genes is rather small with only P300 and NANOG left as gene hubs, we decided to remove this network from the figure and describe the genes in the result section only.

In addition, the reference to feedback loops later in the paragraph is referring to the reciprocal effect of *TBX2* and *MYCN* on their expression levels, which is in this case not related to MYC levels.

To clarify, we adapted the figure accordingly and the text as follows:

“Using iRegulon analysis on the enforced affected genes, motif enrichment of DREAM complex core members, such as FOXM1, E2F4 and MYBL2 was observed in the additively downregulated genes, while ~~motif enrichment~~ ChIP-seq targets for EP300 ~~and~~, REST, NANOG, MYC, and CTBP2 (public datasets) ~~were~~ was found enriched in the upregulated genes (Supplementary Fig. 6d, Supplementary Table 4). “

11. The text refers to a pro-apoptotic effect of combining THZ1 with the HDAC1 inhibitor panobinostat with a call-out to Fig. 7b (p. 13), which only shows effects on cell proliferation (not death). Annexin V staining results suggestive of apoptosis are only shown for the THZ1/JQ1 combination, and only in Supplementary Fig. 7b.

Authors Response: The reviewer has noted this correctly. Indeed, Annexin V staining results upon THZ1 and Panobinostat treatment were originally present in the manuscript, but later removed, without adapting the text accordingly. We added the Annexin V staining results again to the manuscript, see Supplemental Fig. 7e, and adapted the text on page 13 as follows:

“~~Therefore, we first combined the HDAC1 inhibitor panobinostat together with the CDK7 inhibitor THZ1, the latter of which was previously shown to affect transcription of lineage-dependency genes in NB¹ (Fig. 7a) and observed a significantly synergistic effect on cell proliferation and apoptosis in four out of eight NB cell lines (Fig. 7b).”~~

“Given that TBX2 was previously implicated in HDAC1 controlled repression of CDKN1A expression and cell cycle arrest in different cancer types¹⁰ and the strong observed effects of HDAC inhibitors in combination with other anti-cancer drugs¹¹, we also decided to combine HDAC1 inhibitor panobinostat together with the CDK7 inhibitor THZ1, and observed a significantly synergistic effect over time on cell proliferation and apoptosis albeit only in four out of eight NB cell lines (Supplementary Fig 7d-e).”

Further, in addition to the questions from the reviewers, we did some minor changes in the manuscript.

- 1) On page 6 and page 33 we adjusted “Depuydt et al., *JNCI*, in press” to reference number and added the article to the reference list, as the article is published in *JNCI* since March 2018.
- 2) On page 16 we adjusted “Vanhouwaert et al., in preparation; de Carvalho Nunes et al., in preparation” to “unpublished data”.
- 3) Although not questioned by the reviewers, during rebuttal we noticed a minor discrepancy in the text on p. 7 and adapted the text where we changed the numbers in the right order:

“In total, 81 %, 28 % and 94 % of TBX2 binding sites in IMR-32 respectively overlap H3K27ac, H3K4me3 and ATAC-sequencing peaks.”

- 4) We used the IMR-5/75 shMYCN cell line for a lot of experiments in our manuscript, not only for knockdown of MYCN. To avoid misunderstanding, we made the annotation of IMR-5/75 shMYCN cell line more clear in the M&M of the manuscript (see M&M page 21).
- 5) In accordance to the nature publishing group checklist for statistics, we adjusted fig. 5a, 5c and supplementary fig. 5a as individual points need to be shown when there are less than 5 biological replicates, and parametric statistical testing is not justified in case of 3 pairs of biological replicates as sample size is too limited for the normal distribution assumption. That's why we performed a non-parametric Mann-Whitney test and compared control with all shRNAs, which is justified.

For correlation analysis, we initially used the non-parametric test Pearson. We are working with a high sample size which indicates normal distribution according to the central limit theory, and the assumptions "each variable has related pairs", "each variable is continuous" and "absence of outliers" are met, indicating that we can use indeed a non-parametric test. However, MYCN amplification and expression skews the data in some of the datasets, resulting in heteroscedasticity and non-linearity (Levene's Test for Homogeneity of Variance (center = median) is significant ($p < 0.05$)). For this reason, we performed a non-parametric Spearman correlation analysis, and adjusted the R values accordingly in figures supplementary fig. 2d,3f,4e,4f,6a and 6b.

In addition, we adapted the following text to the section "quantification and statistical analysis":

"Statistical significance of differences between conditions for the functional analysis colony forming assay and cell cycle was determined by a non-parametric Mann Whitney test two-sided Student's t test using R package (version 3.3.0) upon mean-centering the datapoints. The ANOVA (analysis of variance) test is used to assess how much of the variability in the TBX2 expression levels can be explained by the patient stage or/and TBX2 copy number status, while the non-parametric test Kruskal-Wallis followed by a post-hoc Dunn's multiple comparisons test was used to determine differences in gene expression and signatures scores between 4 different groups with 4 biological replicates per condition. Statistical significance of overlap between conditions was determined by Fisher test using R package. The non-parametric Spearman or parametric Pearson test was used for correlation analysis depending on the homoscedasticity assumption (Pearson if the assumption is met). Kaplan-meier analysis with log-rank statistics was used for survival analysis. All assumptions for statistical analysis (performed using R) are met and as such justified. For qPCR experiments, reference genes were excluded if Genorm M value was greater than 5 and/or Coefficient of Variation greater than 2, according to the qBaseplus software. For all experiments, at least 3 reference genes were used for the normalisation according to good qPCR practice. The details of quantification and statistical methods used can be found in each figure legend."

- 6) Eight extra co-authors were added to the author list for their contribution to new experiments described in the manuscript and this rebuttal letter.

References:

1. Chipumuro, E. *et al.* CDK7 Inhibition Suppresses Super-Enhancer-Linked Oncogenic Transcription in MYCN-Driven Cancer. *Cell* **159**, 1126–1139 (2014).
2. Kwiatkowski, N. *et al.* Targeting transcription regulation in cancer with a covalent CDK7 inhibitor. *Nature* **511**, 616–620 (2014).
3. Christensen, C. L. *et al.* Targeting Transcriptional Addictions in Small Cell Lung Cancer with a Covalent CDK7 Inhibitor. *Cancer Cell* **26**, 909–922 (2014).
4. Puissant, A. *et al.* Targeting MYCN in Neuroblastoma by BET Bromodomain Inhibition. *Cancer Discov* **3(3)**, 308-23 (2013).
5. Condorelli, F., Gnemmi, I., Vallario, A., Genazzani, A. A. & Canonico, P. L. Inhibitors of histone deacetylase (HDAC) restore the p53 pathway in neuroblastoma cells. *British Journal of Pharmacology* **153**, 657–668 (2008).
6. Kalan, S. *et al.* Activation of the p53 Transcriptional Program Sensitizes Cancer Cells to Cdk7 Inhibitors. *CellReports* **21**, 467–481 (2017).
7. Prince, S., Carreira, S., Vance, K. W., Abrahams, A. & Goding, C. R. Tbx2 Directly Represses the Expression of the p21 WAF1/Cyclin-Dependent Kinase Inhibitor. *Cancer Research* **64**, 1669–1674 (2004).
8. Weiss, W. A., Aldape, K., Mohapatra, G., Feuerstein, B. G. & Bishop, J. M. Targeted expression of MYCN causes neuroblastoma in transgenic mice. *The EMBO Journal* **16**, 2985–2995 (1997).
9. Verfaillie, A., Imrichová, H., Janky, R. & Aerts, S. *iRegulon and i-cisTarget: Reconstructing Regulatory Networks Using Motif and Track Enrichment*. **26**, 2.16.1–2.16.39 (John Wiley & Sons, Inc., 2002).
10. Vance, K. W., Carreira, S., Brosch, G. & Goding, C. R. Tbx2 Is Overexpressed and Plays an Important Role in Maintaining Proliferation and Suppression of Senescence in Melanomas. *Cancer Research* **65**, 2260–2268 (2005).
11. Eckschlager, T., Plch, J., Stiborova, M. & Hrabeta, J. Histone Deacetylase Inhibitors as Anticancer Drugs. *IJMS* **18**, 1414 (2017).

REVIEWERS' COMMENTS:

Reviewer #1 (Remarks to the Author):

The authors have expanded their preclinical studies with new organoid testing which does widen the scope of these findings, however this is still not sufficient for potential translation of this hypothesis to the clinic. As a basic science manuscript, I feel the authors have sufficiently addressed the reviewer comments.

Reviewer #2 (Remarks to the Author):

This is a revised version of a manuscript I reviewed previously. The authors have responded thoughtfully and constructively to my concerns and those of the other reviewer, and greatly strengthened the paper. It is now suitable for publication without further review, in my opinion.

We thank the reviewers for the positive feedback and we are thankful for the substantial amount of time the reviewers spend to review this manuscript

REVIEWERS' COMMENTS:

Reviewer #1 (Remarks to the Author):

The authors have expanded their preclinical studies with new organoid testing which does widen the scope of these findings, however this is still not sufficient for potential translation of this hypothesis to the clinic. As a basic science manuscript, I feel the authors have sufficiently addressed the reviewer comments.

Reviewer #2 (Remarks to the Author):

This is a revised version of a manuscript I reviewed previously. The authors have responded thoughtfully and constructively to my concerns and those of the other reviewer, and greatly strengthened the paper. It is now suitable for publication without further review, in my opinion.